# Robust Estimation and Generative Adversarial Networks

**Chao Gao**
Department of Statistics
University of Chicago
Chicago, IL 60637 USA
chaogao@galton.uchicago.edu

**Jiyi Liu**
Department of Statistics and Data Science
Yale University
New Haven, CT 06511 USA
jiyi.liu@yale.edu

**Yuan Yao & Weizhi Zhu**
Department of Mathematics
Hong Kong University of Science and Technology
Kowloon, Hong Kong
yuany@ust.hk; wzhuai@connect.ust.hk

## Abstract

Robust estimation under Huber's $\epsilon$-contamination model has become an important topic in statistics and theoretical computer science. Statistically optimal procedures such as Tukey's median and other estimators based on depth functions are impractical because of their computational intractability. In this paper, we establish an intriguing connection between $f$-GANs and various depth functions through the lens of $f$-Learning. Similar to the derivation of $f$-GANs, we show that these depth functions that lead to statistically optimal robust estimators can all be viewed as variational lower bounds of the total variation distance in the framework of $f$-Learning. This connection opens the door of computing robust estimators using tools developed for training GANs. In particular, we show in both theory and experiments that some appropriate structures of discriminator networks with hidden layers in GANs lead to statistically optimal robust location estimators for both Gaussian distribution and general elliptical distributions where first moment may not exist.

## 1 Introduction

In the setting of Huber's $\epsilon$-contamination model (Huber, 1964; 1965), one has i.i.d observations

$$X_1, ..., X_n \sim (1 - \epsilon)P_\theta + \epsilon Q, \tag{1}$$

and the goal is to estimate the model parameter $\theta$. Under the data generating process (1), each observation has a $1 - \epsilon$ probability to be drawn from $P_\theta$ and the other $\epsilon$ probability to be drawn from the contamination distribution $Q$. The presence of an unknown contamination distribution poses both statistical and computational challenges to the problem. For example, consider a normal mean estimation problem with $P_\theta = N(\theta, I_p)$. Due to the contamination of data, the sample average, which is optimal when $\epsilon = 0$, can be arbitrarily far away from the true mean if $Q$ charges a positive probability at infinity. Moreover, even robust estimators such as coordinatewise median and geometric median are proved to be suboptimal under the setting of (1) (Chen et al., 2018; Diakonikolas et al., 2016a; Lai et al., 2016). The search for both statistically optimal and computationally feasible procedures has become a fundamental problem in areas including statistics and computer science.

For the normal mean estimation problem, it has been shown in Chen et al. (2018) that the minimax rate with respect to the squared $\ell_2$ loss is $\frac{p}{n} \vee \epsilon^2$, and is achieved by Tukey's median (Tukey, 1975). Despite the statistical optimality of Tukey's median, its computation is not tractable. In fact, even an approximate algorithm takes $O(e^{Cp})$ in time (Amenta et al., 2000; Chan, 2004; Rousseeuw & Struyf, 1998).

Recent developments in theoretical computer science are focused on the search of computationally tractable algorithms for estimating $\theta$ under Huber's $\epsilon$-contamination model (1). The success of the efforts started from two fundamental papers Diakonikolas et al. (2016a); Lai et al. (2016), where two different but related computational strategies "iterative filtering" and "dimension halving" were proposed to robustly estimate the normal mean. These algorithms can provably achieve the minimax rate $\frac{p}{n} \vee \epsilon^2$ up to a poly-logarithmic factor in polynomial time. The main idea behind the two methods is a critical fact that a good robust moment estimator can be certified efficiently by higher moments. This idea was later further extended (Diakonikolas et al., 2017; Du et al., 2017; Diakonikolas et al., 2016b; 2018a;c;b; Kothari et al., 2018) to develop robust and computable procedures for various other problems.

However, many of the computationally feasible procedures for robust mean estimation in the literature rely on the knowledge of covariance matrix and sometimes the knowledge of contamination proportion. Even though these assumptions can be relaxed, nontrivial modifications of the algorithms are required for such extensions and statistical error rates may also be affected. Compared with these computationally feasible procedures proposed in the recent literature for robust estimation, Tukey's median (9) and other depth-based estimators (Rousseeuw & Hubert, 1999; Mizera, 2002; Zhang, 2002; Mizera & Müller, 2004; Paindaveine & Van Bever, 2017) have some indispensable advantages in terms of their statistical properties. First, the depth-based estimators have clear objective functions that can be interpreted from the perspective of projection pursuit (Mizera, 2002). Second, the depth-based procedures are adaptive to unknown nuisance parameters in the models such as covariance structures, contamination proportion, and error distributions (Chen et al., 2018; Gao, 2017). Last but not least, Tukey's depth and other depth functions are mostly designed for robust quantile estimation, while the recent advancements in the theoretical computer science literature are all focused on robust moments estimation. Although this is not an issue when it comes to normal mean estimation, the difference is fundamental for robust estimation under general settings such as elliptical distributions where moments do not necessarily exist.

Given the desirable statistical properties discussed above, this paper is focused on the development of computational strategies of depth-like procedures. Our key observation is that robust estimators that are maximizers of depth functions, including halfspace depth, regression depth and covariance matrix depth, can all be derived under the framework of $f$-GAN (Nowozin et al., 2016). As a result, these depth-based estimators can be viewed as minimizers of variational lower bounds of the total variation distance between the empirical measure and the model distribution (Proposition 2.1). This observation allows us to leverage the recent developments in the deep learning literature to compute these variational lower bounds through neural network approximations. Our theoretical results give insights on how to choose appropriate neural network classes that lead to minimax optimal robust estimation under Huber's $\epsilon$-contamination model. In particular, Theorem 3.1 and 3.2 characterize the networks which can robustly estimate the Gaussian mean by TV-GAN and JS-GAN, respectively; Theorem 4.1 is an extension to robust location estimation under the class of elliptical distributions which includes Cauchy distribution whose mean does not exist. Numerical experiments in Section 5 are provided to show the success of these GANs.

## 2 ROBUST ESTIMATION AND $f$-GAN

We start with the definition of $f$-divergence (Csiszár, 1964; Ali & Silvey, 1966). Given a strictly convex function $f$ that satisfies $f(1) = 0$, the $f$-GAN between two probability distributions $P$ and $Q$ is defined by

$$D_f(P\|Q) = \int f\left(\frac{p}{q}\right) dQ. \tag{2}$$

Here, we use $p(\cdot)$ and $q(\cdot)$ to stand for the density functions of $P$ and $Q$ with respect to some common dominating measure. For a fully rigorous definition, see Polyanskiy & Wu (2017). Let $f^*$ be the convex conjugate of $f$. That is, $f^*(t) = \sup_{u \in \mathbf{dom}_f}(ut - f(u))$. A variational lower bound of (2) is

$$D_f(P\|Q) \geq \sup_{T \in \mathcal{T}} \left[E_P T(X) - E_Q f^*(T(X))\right]. \tag{3}$$

Note that the inequality (3) holds for any class $\mathcal{T}$, and it becomes an equality whenever the class $\mathcal{T}$ contains the function $f'(p/q)$ (Nguyen et al., 2010). For notational simplicity, we also use $f'$ for an arbitrary element of

the subdifferential when the derivative does not exist. With i.i.d. observations $X_1, ..., X_n \sim P$, the variational lower bound (3) naturally leads to the following learning method

$$\widehat{P} = \underset{Q \in \mathcal{Q}}{\operatorname{argmin}} \, \underset{T \in \mathcal{T}}{\sup} \left[ \frac{1}{n} \sum_{i=1}^{n} T(X_i) - E_Q f^*(T(X)) \right]. \tag{4}$$

The formula (4) is a powerful and general way to learn the distribution $P$ from its i.i.d. observations. It is known as $f$-GAN (Nowozin et al., 2016), an extension of GAN (Goodfellow et al., 2014), which stands for *generative adversarial networks*. The idea is to find a $\widehat{P}$ so that the best discriminator $T$ in the class $\mathcal{T}$ cannot tell the difference between $\widehat{P}$ and the empirical distribution $\frac{1}{n} \sum_{i=1}^{n} \delta_{X_i}$.

## 2.1 $f$-LEARNING: A UNIFIED FRAMEWORK

Our $f$-Learning framework is based on a special case of the variational lower bound (3). That is,

$$D_f(P\|Q) \geq \underset{\widetilde{Q} \in \widetilde{\mathcal{Q}}_Q}{\sup} \left[ E_P f'\left(\frac{\widetilde{q}(X)}{q(X)}\right) - E_Q f^*\left(f'\left(\frac{\widetilde{q}(X)}{q(X)}\right)\right) \right], \tag{5}$$

where $\widetilde{q}(\cdot)$ stands for the density function of $\widetilde{Q}$. Note that here we allow the class $\widetilde{\mathcal{Q}}_Q$ to depend on the distribution $Q$ in the second argument of $D_f(P\|Q)$. Compare (5) with (3), and it is easy to realize that (5) is a special case of (3) with

$$\mathcal{T} = \mathcal{T}_Q = \left\{ f'\left(\frac{\widetilde{q}}{q}\right) : \widetilde{q} \in \widetilde{\mathcal{Q}}_Q \right\}. \tag{6}$$

Moreover, the inequality (5) becomes an equality as long as $P \in \widetilde{\mathcal{Q}}_Q$. The sample version of (5) leads to the following learning method

$$\widehat{P} = \underset{Q \in \mathcal{Q}}{\operatorname{argmin}} \, \underset{\widetilde{Q} \in \widetilde{\mathcal{Q}}_Q}{\sup} \left[ \frac{1}{n} \sum_{i=1}^{n} f'\left(\frac{\widetilde{q}(X_i)}{q(X_i)}\right) - E_Q f^*\left(f'\left(\frac{\widetilde{q}(X)}{q(X)}\right)\right) \right]. \tag{7}$$

The learning method (7) will be referred to as $f$-Learning in the sequel. It is a very general framework that covers many important learning procedures as special cases. For example, consider the special case where $\widetilde{\mathcal{Q}}_Q = \widetilde{\mathcal{Q}}$ independent of $Q$, $\mathcal{Q} = \widetilde{\mathcal{Q}}$, and $f(x) = x \log x$. Direct calculations give $f'(x) = \log x + 1$ and $f^*(t) = e^{t-1}$. Therefore, (7) becomes

$$\widehat{P} = \underset{Q \in \mathcal{Q}}{\operatorname{argmin}} \, \underset{\widetilde{Q} \in \mathcal{Q}}{\sup} \frac{1}{n} \sum_{i=1}^{n} \log \frac{\widetilde{q}(X_i)}{q(X_i)} = \underset{q \in \mathcal{Q}}{\operatorname{argmax}} \frac{1}{n} \sum_{i=1}^{n} \log q(X_i),$$

which is the *maximum likelihood estimator* (MLE).

## 2.2 TV-LEARNING AND DEPTH-BASED ESTIMATORS

An important generator $f$ that we will discuss here is $f(x) = (x-1)_+$. This leads to the total variation distance $D_f(P\|Q) = \frac{1}{2} \int |p - q|$. With $f'(x) = \mathbb{I}\{x \geq 1\}$ and $f^*(t) = t\mathbb{I}\{0 \leq t \leq 1\}$, the *TV-Learning* is given by

$$\widehat{P} = \underset{Q \in \mathcal{Q}}{\operatorname{argmin}} \, \underset{\widetilde{Q} \in \mathcal{Q}_Q}{\sup} \left[ \frac{1}{n} \sum_{i=1}^{n} \mathbb{I}\left\{ \frac{\widetilde{q}(X_i)}{q(X_i)} \geq 1 \right\} - Q\left( \frac{\widetilde{q}}{q} \geq 1 \right) \right]. \tag{8}$$

A closely related idea was previously explored by Yatracos (1985); Devroye & Lugosi (2012). The following proposition shows that when $\widetilde{\mathcal{Q}}_Q$ approaches to $\mathcal{Q}$ in some neighborhood, TV-Learning leads to robust estimators that are defined as the maximizers of various depth functions including Tukey's depth, regression depth, and covariance depth.

**Proposition 2.1.** *The TV-Learning (8) includes the following special cases:*

1. *Tukey's halfspace depth: Take* $\mathcal{Q} = \{N(\eta, I_p) : \eta \in \mathbb{R}^p\}$ *and* $\widetilde{\mathcal{Q}}_\eta = \{N(\widetilde{\eta}, I_p) : \|\widetilde{\eta} - \eta\| \leq r\}$. *As* $r \to 0$, *(8) becomes*

$$\widehat{\theta} = \underset{\eta \in \mathbb{R}^p}{\mathrm{argmax}} \inf_{\|u\|=1} \frac{1}{n} \sum_{i=1}^n \mathbb{I}\{u^T(X_i - \eta) \geq 0\}. \tag{9}$$

2. *Regression depth: Take* $\mathcal{Q} = \left\{ P_{y,X} = P_{y|X} P_X : P_{y|X} = N(X^T\eta, 1), \eta \in \mathbb{R}^p \right\}$,
   *and* $\widetilde{\mathcal{Q}}_\eta = \left\{ P_{y,X} = P_{y|X} P_X : P_{y|X} = N(X^T\widetilde{\eta}, 1), \|\widetilde{\eta} - \eta\| \leq r \right\}$. *As* $r \to 0$, *(8) becomes*

$$\widehat{\theta} = \underset{\eta \in \mathbb{R}^p}{\mathrm{argmax}} \inf_{\|u\|=1} \frac{1}{n} \sum_{i=1}^n \mathbb{I}\{u^T X_i(y_i - X_i^T\eta) \geq 0\}. \tag{10}$$

3. *Covariance matrix depth: Take* $\mathcal{Q} = \{N(0, \Gamma) : \Gamma \in \mathcal{E}_p\}$, *where* $\mathcal{E}_p$ *stands for the class of* $p \times p$ *covariance matrices, and* $\widetilde{\mathcal{Q}}_\Gamma = \left\{ N(0, \widetilde{\Gamma}) : \widetilde{\Gamma}^{-1} = \Gamma^{-1} + \widetilde{r}uu^T \in \mathcal{E}_p, |\widetilde{r}| \leq r, \|u\| = 1 \right\}$. *As* $r \to 0$, *(8) becomes*

$$\widehat{\Sigma} = \underset{\Gamma \in \mathcal{E}_p}{\mathrm{argmin}} \sup_{\|u\|=1} \left[ \left( \frac{1}{n} \sum_{i=1}^n \mathbb{I}\{|u^T X_i|^2 \leq u^T\Gamma u\} - \mathbb{P}(\chi_1^2 \leq 1) \right) \right. \tag{11}$$
$$\left. \vee \left( \frac{1}{n} \sum_{i=1}^n \mathbb{I}\{|u^T X_i|^2 > u^T\Gamma u\} - \mathbb{P}(\chi_1^2 > 1) \right) \right].$$

The formula (9) is recognized as *Tukey's median*, the maximizer of Tukey's halfspace depth. A traditional understanding of Tukey's median is that (9) maximizes the halfspace depth (Donoho & Gasko, 1992) so that $\widehat{\theta}$ is close to the centers of all one-dimensional projections of the data. In the $f$-Learning framework, $N(\widehat{\theta}, I_p)$ is understood to be the minimizer of a variational lower bound of the total variation distance. The formula (10) gives the estimator that maximizes the *regression depth* proposed by Rousseeuw & Hubert (1999). It is worth noting that the derivation of (10) does not depend on the marginal distribution $P_X$ in the linear regression model. Finally, (11) is related to the *covariance matrix depth* (Zhang, 2002; Chen et al., 2018; Paindaveine & Van Bever, 2017). All of the estimators (9), (10) and (11) are proved to achieve the minimax rate for the corresponding problems under Huber's $\epsilon$-contamination model (Chen et al., 2018; Gao, 2017).

## 2.3 FROM $f$-LEARNING TO $f$-GAN

The connection to various depth functions shows the importance of TV-Learning in robust estimation. However, it is well-known that depth-based estimators are very hard to compute (Amenta et al., 2000; van Kreveld et al., 1999; Rousseeuw & Struyf, 1998), which limits their applications only for very low-dimensional problems. On the other hand, the general $f$-GAN framework (4) has been successfully applied to learn complex distributions and images in practice (Goodfellow et al., 2014; Radford et al., 2015; Salimans et al., 2016). The major difference that gives the computational advantage to $f$-GAN is its flexibility in terms of designing the discriminator class $\mathcal{T}$ using neural networks compared with the pre-specified choice (6) in $f$-Learning. While $f$-Learning provides a unified perspective in understanding various depth-based procedures in robust estimation, we can step back into the more general $f$-GAN for its computational advantages, and to design efficient computational strategies.

## 3 ROBUST MEAN ESTIMATION VIA GAN

In this section, we focus on the problem of robust mean estimation under Huber's $\epsilon$-contamination model. Our goal is to reveal how the choice of the class of discriminators affects robustness and statistical optimality under the simplest possible setting. That is, we have i.i.d. observations $X_1, ..., X_n \sim (1 - \epsilon)N(\theta, I_p) + \epsilon Q$, and we need to estimate the unknown location $\theta \in \mathbb{R}^p$ with the contaminated data. Our goal is to achieve the minimax rate $\frac{p}{n} \vee \epsilon^2$ with respect to the squared $\ell_2$ loss uniformly over all $\theta \in \mathbb{R}^p$ and all $Q$.

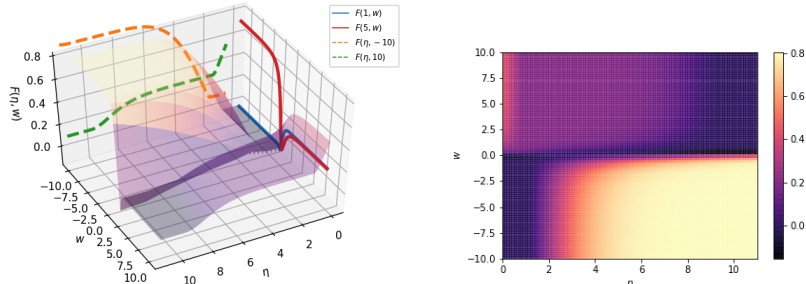

Figure 1: Landscape of TV-GAN objective function $F(\eta, w) = \sup_b[E_P\text{sigmoid}(wX + b) - E_{N(\eta,1)}\text{sigmoid}(wX + b)]$, where $b$ is maximized out for visualization. Samples are drawn from $P = (1 - \epsilon)N(1, 1) + \epsilon N(10, 1)$ with $\epsilon = 0.2$. Left: a surface plot of $F(\eta, w)$. The solid curves are marginal functions for fixed $\eta$'s: $F(1, w)$ (red) and $F(5, w)$ (blue), and the dash curves are marginal functions for fixed $w$'s: $F(\eta, -10)$ (orange) and $F(\eta, 10)$ (green). Right: a heatmap of $F(\eta, w)$. It is clear that $\tilde{F}(w) = F(\eta, w)$ has two local maxima for a given $\eta$, achieved at $w = +\infty$ and $w = -\infty$. In fact, the global maximum for $\tilde{F}(w)$ has a phase transition from $w = +\infty$ to $w = -\infty$ as $\eta$ grows. For example, the maximum is achieved at $w = +\infty$ when $\eta = 1$ (blue solid) and is achieved at $w = -\infty$ when $\eta = 5$ (red solid). Unfortunately, even if we initialize with $\eta_0 = 1$ and $w_0 > 0$, gradient ascents on $\eta$ will only increase the value of $\eta$ (green dash), and thus as long as the discriminator cannot reach the global maximizer, $w$ will be stuck in the positive half space $\{w : w > 0\}$ and further increase the value of $\eta$.

## 3.1 RESULTS FOR TV-GAN

We start with the total variation GAN (TV-GAN) with $f(x) = (x - 1)_+$ in (4). For the Gaussian location family, (4) can be written as

$$\widehat{\theta} = \underset{\eta \in \mathbb{R}^p}{\arg\min} \max_{D \in \mathcal{D}} \left[\frac{1}{n} \sum_{i=1}^n D(X_i) - E_{N(\eta, I_p)}D(X)\right], \tag{12}$$

with $T(x) = D(x)$ in (4). Now we need to specify the class of discriminators $\mathcal{D}$ to solve the classification problem between $N(\eta, I_p)$ and the empirical distribution $\frac{1}{n}\sum_{i=1}^n \delta_{X_i}$. One of the simplest discriminator classes is the logistic regression,

$$\mathcal{D} = \left\{D(x) = \text{sigmoid}(w^T x + b) : w \in \mathbb{R}^p, b \in \mathbb{R}\right\}. \tag{13}$$

With $D(x) = \text{sigmoid}(w^T x + b) = (1 + e^{-w^T x - b})^{-1}$ in (13), the procedure (12) can be viewed as a smoothed version of TV-Learning (8). To be specific, the sigmoid function $\text{sigmoid}(w^T x + b)$ tends to an indicator function as $\|w\| \to \infty$, which leads to a procedure very similar to (9). In fact, the class (13) is richer than the one used in (9), and thus (12) can be understood as the minimizer of a sharper variational lower bound than that of (9).

**Theorem 3.1.** *Assume $\frac{p}{n} + \epsilon^2 \leq c$ for some sufficiently small constant $c > 0$. With i.i.d. observations $X_1, ..., X_n \sim (1 - \epsilon)N(\theta, I_p) + \epsilon Q$, the estimator $\widehat{\theta}$ defined by (12) satisfies*

$$\|\widehat{\theta} - \theta\|^2 \leq C\left(\frac{p}{n} \vee \epsilon^2\right),$$

*with probability at least $1 - e^{-C'(p + n\epsilon^2)}$ uniformly over all $\theta \in \mathbb{R}^p$ and all $Q$. The constants $C, C' > 0$ are universal.*

Though TV-GAN can achieve the minimax rate $\frac{p}{n} \vee \epsilon^2$ under Huber's contamination model, it may suffer from optimization difficulties especially when the distributions $Q$ and $N(\theta, I_p)$ are far away from each other, as shown in Figure 1.

## 3.2 RESULTS FOR JS-GAN

Given the intractable optimization property of TV-GAN, we next turn to Jensen-Shannon GAN (JS-GAN) with $f(x) = x \log x - (x+1) \log \frac{x+1}{2}$. The estimator is defined by

$$\widehat{\theta} = \underset{\eta \in \mathbb{R}^p}{\operatorname{argmin}} \max_{D \in \mathcal{D}} \left[ \frac{1}{n} \sum_{i=1}^{n} \log D(X_i) + E_{N(\eta, I_p)} \log(1 - D(X_i)) \right] + \log 4, \tag{14}$$

with $T(x) = \log D(x)$ in (4). This is exactly the original GAN (Goodfellow et al., 2014) specialized to the normal mean estimation problem. The advantages of JS-GAN over other forms of GAN have been studied extensively in the literature (Lucic et al., 2017; Kurach et al., 2018).

Unlike TV-GAN, our experiment results show that (14) with the logistic regression discriminator class (13) is not robust to contamination. However, if we replace (13) by a neural network class with one or more hidden layers, the estimator will be robust and will also work very well numerically.

To understand why and how the class of the discriminators affects the robustness property of JS-GAN, we introduce a new concept called restricted Jensen-Shannon divergence. Let $g : \mathbb{R}^p \to \mathbb{R}^d$ be a function that maps a $p$-dimensional observation to a $d$-dimensional feature space. The restricted Jensen-Shannon divergence between two probability distributions $P$ and $Q$ with respect to the feature $g$ is defined as

$$\mathsf{JS}_g(P, Q) = \max_{w \in \mathcal{W}} \left[ E_P \log \mathsf{sigmoid}(w^T g(X)) + E_Q \log(1 - \mathsf{sigmoid}(w^T g(X))) \right] + \log 4.$$

In other words, $P$ and $Q$ are distinguished by a logistic regression classifier that uses the feature $g(X)$. It is easy to see that $\mathsf{JS}_g(P, Q)$ is a variational lower bound of the original Jensen-Shannon divergence. The key property of $\mathsf{JS}_g(P, Q)$ is given by the following proposition.

**Proposition 3.1.** *Assume $\mathcal{W}$ is a convex set that contains an open neighborhood of $0$. Then, $\mathsf{JS}_g(P, Q) = 0$ if and only if $E_P g(X) = E_Q g(X)$.*

The proposition asserts that $\mathsf{JS}_g(\cdot, \cdot)$ cannot distinguish $P$ and $Q$ if the feature $g(X)$ has the same expected value under the two distributions. This *generalized moment matching effect* has also been studied by Liu et al. (2017) for general $f$-GANs. However, the linear discriminator class considered in Liu et al. (2017) is parameterized in a different way compared with the discriminator class here.

When we apply Proposition 3.1 to robust mean estimation, the JS-GAN is trying to match the values of $\frac{1}{n} \sum_{i=1}^{n} g(X_i)$ and $E_{N(\eta, I_n)} g(X)$ for the feature $g(X)$ used in the logistic regression classifier. This explains what we observed in our numerical experiments. A neural net without any hidden layer is equivalent to a logistic regression with a linear feature $g(X) = (X^T, 1)^T \in \mathbb{R}^{p+1}$. Therefore, whenever $\eta = \frac{1}{n} \sum_{i=1}^{n} X_i$, we have $\mathsf{JS}_g\left(\frac{1}{n} \sum_{i=1}^{n} \delta_{X_i}, N(\eta, I_p)\right) = 0$, which implies that the sample mean is a global maximizer of (14). On the other hand, a neural net with at least one hidden layers involves a nonlinear feature function $g(X)$, which is the key that leads to the robustness of (14).

We will show rigorously that a neural net with one hidden layer is sufficient to make (14) robust and optimal. Consider the following class of discriminators,

$$\mathcal{D} = \left\{ D(x) = \mathsf{sigmoid} \left( \sum_{j \geq 1} w_j \sigma(u_j^T x + b_j) \right) : \sum_{j \geq 1} |w_j| \leq \kappa, u_j \in \mathbb{R}^p, b_j \in \mathbb{R} \right\}. \tag{15}$$

The class (15) consists of two-layer neural network functions. While the dimension of the input layer is $p$, the dimension of the hidden layer can be arbitrary, as long as the weights have a bounded $\ell_1$ norm. The nonlinear activation function $\sigma(\cdot)$ is allowed to take 1) indicator: $\sigma(x) = \mathbb{I}\{x \geq 1\}$, 2) sigmoid: $\sigma(x) = \frac{1}{1+e^{-x}}$, 3) ramp: $\sigma(x) = \max(\min(x + 1/2, 1), 0)$. Other bounded activation functions are also possible, but we do not exclusively list them. The rectified linear unit (ReLU) will be studied in Appendix A.

**Theorem 3.2.** *Consider the estimator $\widehat{\theta}$ defined by (14) with $\mathcal{D}$ specified by (15). Assume $\frac{p}{n} + \epsilon^2 \leq c$ for some sufficiently small constant $c > 0$, and set $\kappa = O\left(\sqrt{\frac{p}{n}} + \epsilon\right)$. With i.i.d. observations $X_1, ..., X_n \sim$*

$(1 - \epsilon)N(\theta, I_p) + \epsilon Q$, we have

$$\|\widehat{\theta} - \theta\|^2 \leq C \left( \frac{p}{n} \vee \epsilon^2 \right),$$

with probability at least $1 - e^{-C'(p+n\epsilon^2)}$ uniformly over all $\theta \in \mathbb{R}^p$ and all $Q$. The constants $C, C' > 0$ are universal.

## 4 ELLIPTICAL DISTRIBUTIONS

An advantage of Tukey's median (9) is that it leads to optimal robust location estimation under general elliptical distributions such as Cauchy distribution whose mean does not exist. In this section, we show that JS-GAN shares the same property. A random vector $X \in \mathbb{R}^p$ follows an elliptical distribution if it admits a representation

$$X = \theta + \xi A U,$$

where $U$ is uniformly distributed on the unit sphere $\{u \in \mathbb{R}^p : \|u\| = 1\}$ and $\xi \geq 0$ is a random variable independent of $U$ that determines the shape of the elliptical distribution (Fang, 2017). The center and the scatter matrix is $\theta$ and $\Sigma = AA^T$.

For a unit vector $v$, let the density function of $\xi v^T U$ be $h$. Note that $h$ is independent of $v$ because of the symmetry of $U$. Then, there is a one-to-one relation between the distribution of $\xi$ and $h$, and thus the triplet $(\theta, \Sigma, h)$ fully parametrizes an elliptical distribution.

Note that $h$ and $\Sigma = AA^T$ are not identifiable, because $\xi A = (c\xi)(c^{-1}A)$ for any $c > 0$. Therefore, without loss of generality, we can restrict $h$ to be a member of the following class

$$\mathcal{H} = \left\{ h : h(t) = h(-t), h \geq 0, \int h = 1, \int \sigma(t)(1 - \sigma(t))h(t)dt = 1 \right\}.$$

This makes the parametrization $(\theta, \Sigma, h)$ of an elliptical distribution fully identifiable, and we use $EC(\theta, \Sigma, h)$ to denote an elliptical distribution parametrized in this way.

The JS-GAN estimator is defined as

$$(\widehat{\theta}, \widehat{\Sigma}, \widehat{h}) = \operatorname*{argmin}_{\eta \in \mathbb{R}^p, \Gamma \in \mathcal{E}_p(M), g \in \mathcal{H}} \max_{D \in \mathcal{D}} \left[ \frac{1}{n} \sum_{i=1}^{n} \log D(X_i) + E_{EC(\eta, \Gamma, g)} \log(1 - D(X)) \right] + \log 4, \qquad (16)$$

where $\mathcal{E}_p(M)$ is the set of all positive semi-definite matrix with spectral norm bounded by $M$.

**Theorem 4.1.** *Consider the estimator $\widehat{\theta}$ defined above with $\mathcal{D}$ specified by (15). Assume $M = O(1)$, $\frac{p}{n} + \epsilon^2 \leq c$ for some sufficiently small constant $c > 0$, and set $\kappa = O\left(\sqrt{\frac{p}{n}} + \epsilon\right)$. With i.i.d. observations $X_1, ..., X_n \sim (1 - \epsilon)EC(\theta, \Sigma, h) + \epsilon Q$, we have*

$$\|\widehat{\theta} - \theta\|^2 \leq C \left( \frac{p}{n} \vee \epsilon^2 \right),$$

*with probability at least $1 - e^{-C'(p+n\epsilon^2)}$ uniformly over all $\theta \in \mathbb{R}^p$, $\Sigma \in \mathcal{E}_p(M)$ and all $Q$. The constants $C, C' > 0$ are universal.*

**Remark 4.1.** *The result of Theorem 4.1 also holds (and is proved) under the strong contamination model (Diakonikolas et al., 2016a). That is, we have i.i.d. observations $X_1, ..., X_n \sim P$ for some $P$ satisfying $\mathsf{TV}(P, EC(\theta, \Sigma, h)) \leq \epsilon$. See its proof in Appendix D.2.*

Note that Theorem 4.1 guarantees the same convergence rate as in the Gaussian case for all elliptical distributions. This even includes multivariate Cauchy where mean does not exist. Therefore, the location estimator (16) is fundamentally different from Diakonikolas et al. (2016a); Lai et al. (2016), which is only designed for robust mean estimation. We will show such a difference in our numerical results.

To achieve rate-optimality for robust location estimation under general elliptical distributions, the estimator (16) is different from (14) only in the generator class. They share the same discriminator class (15). This underlines

an important principle for designing GAN estimators: the overall statistical complexity of the estimator is only determined by the discriminator class.

The estimator (16) also outputs $(\widehat{\Sigma}, \widehat{h})$, but we do not claim any theoretical property for $(\widehat{\Sigma}, \widehat{h})$ in this paper. This will be systematically studied in a future project.

## 5 NUMERICAL EXPERIMENTS

In this section, we give extensive numerical studies of robust mean estimation via GAN. After introducing the implementation details in Section 5.1, we verify our theoretical results on minimax estimation with both TV-GAN and JS-GAN in Section 5.2. Comparison with other methods on robust mean estimation in the literature is given in Section 5.3. The effects of various network structures are studied in Section 5.4. Adaptation to unknown covariance is studied in Section 5.5. In all these cases, we assume i.i.d. observations are drawn from $(1 - \epsilon)N(0_p, I_p) + \epsilon Q$ with $\epsilon$ and $Q$ to be specified. Finally, adaptation to elliptical distributions is studied in Section 5.6.

### 5.1 IMPLEMENTATIONS

We adopt the standard algorithmic framework of $f$-GANs (Nowozin et al., 2016) for the implementation of JS-GAN and TV-GAN for robust mean estimation. In particular, the generator for mean estimation is $G_\eta(Z) = Z + \eta$ with $Z \sim N(0_p, I_p)$; the discriminator $D$ is a multilayer perceptron (MLP), where each layer consisting of a linear map and a sigmoid activation function and the number of nodes will vary in different experiments to be specified below. Details related to algorithms, tuning, critical hyper-parameters, structures of discriminator networks and other training tricks for stabilization and acceleration are discussed in Appendix B.1. A PyTorch implementation is available at https://github.com/zhuwzh/Robust-GAN-Center.

### 5.2 NUMERICAL SUPPORTS FOR THE MINIMAX RATES

We verify the minimax rates achieved by TV-GAN (Theorem 3.1) and JS-GAN (Theorem 3.2) via numerical experiments. Two main scenarios we consider here are $\sqrt{p/n} < \epsilon$ and $\sqrt{p/n} > \epsilon$, where in both cases, various types of contamination distributions $Q$ are considered. Specifically, the choice of contamination distributions $Q$ includes $N(\mu * 1_p, I_p)$ with $\mu$ ranges in $\{0.2, 0.5, 1, 5\}$, $N(0.5 * 1_p, \Sigma)$ and Cauchy$(\tau * 1_p)$. Details of the construction of the covariance matrix $\Sigma$ is given in Appendix B.2. The distribution Cauchy$(\tau * 1_p)$ is obtained by combining $p$ independent one-dimensional standard Cauchy with location parameter $\tau_j = 0.5$.

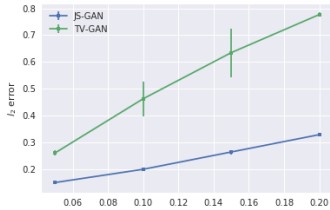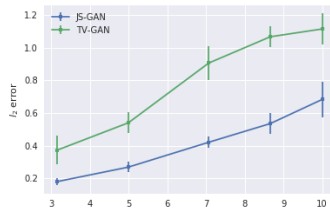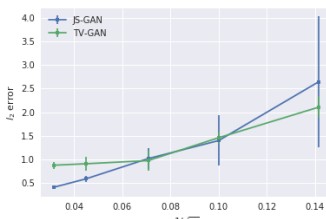

Figure 2: $\ell_2$ error $\|\widehat{\theta} - \theta\|$ against $\epsilon$ (left: $p = 100$, $n = 50,000$ and $\epsilon$ ranges from 0.05 to 0.20), $\sqrt{p}$ (middle: $n = 1,000$, $\epsilon = 0.1$ and $p$ ranges from 10 to 100) and $1/\sqrt{n}$ (right: $p = 50$, $\epsilon = 0.1$ and $n$ ranges from 50 to 1,000), respectively. Net structure: One hidden layer with 20 hidden units (JS-GAN), zero hidden layer (TV-GAN). The vertical bars indicate $\pm$ standard deviations.

The main experimental results are summarized in Figure 2, where the $\ell_2$ error we present is the maximum error among all choices of $Q$, and detailed numerical results can be founded in Tables 7, 8 and 9 in Appendix. We separately explore the relation between the error and one of $\epsilon$, $\sqrt{p}$ and $1/\sqrt{n}$ with the other two parameters fixed. The study of the relation between the $\ell_2$ error and $\epsilon$ is in the regime $\sqrt{p/n} < \epsilon$ so that $\epsilon$ dominates

the minimax rate. The scenario $\sqrt{p/n} > \epsilon$ is considered in the study of the effects of $\sqrt{p}$ and $1/\sqrt{n}$. As is shown in Figure 2, the errors are approximately linear against the corresponding parameters in all cases, which empirically verifies the conclusions of Theorem 3.1 and Theorem 3.2.

## 5.3 COMPARISONS WITH OTHER METHODS

We perform additional experiments to compare with other methods including *dimension halving* (Lai et al., 2016) and *iterative filtering* (Diakonikolas et al., 2017) under various settings. We emphasize that our method does not require any knowledge about the nuisance parameters such as the contamination proportion $\epsilon$. Tuning GAN is only a matter of optimization and one can tune parameters based on the objective function only.

Table 1: Comparison of various robust mean estimation methods. Net structure: One-hidden layer network with 20 hidden units when $n = 50,000$ and 2 hidden units when $n = 5,000$. The number in each cell is the average of $\ell_2$ error $\|\widehat{\theta} - \theta\|$ with standard deviation in parenthesis estimated from 10 repeated experiments and the smallest error among four methods is highlighted in bold.

| $Q$ | $n$ | $p$ | $\epsilon$ | TV-GAN | JS-GAN | Dimension Halving | Iterative Filtering |
|---|---|---|---|---|---|---|---|
| $N(0.5 * 1_p, I_p)$ | 50,000 | 100 | .2 | **0.0953 (0.0064)** | 0.1144 (0.0154) | 0.3247 (0.0058) | 0.1472 (0.0071) |
| $N(0.5 * 1_p, I_p)$ | 5,000 | 100 | .2 | **0.1941 (0.0173)** | 0.2182 (0.0527) | 0.3568 (0.0197) | 0.2285 (0.0103) |
| $N(0.5 * 1_p, I_p)$ | 50,000 | 200 | .2 | **0.1108 (0.0093)** | 0.1573 (0.0815) | 0.3251 (0.0078) | 0.1525 (0.0045) |
| $N(0.5 * 1_p, I_p)$ | 50,000 | 100 | .05 | 0.0913 (0.0527) | 0.1390 (0.0050) | 0.0814 (0.0056) | **0.0530 (0.0052)** |
| $N(5 * 1_p, I_p)$ | 50,000 | 100 | .2 | 2.7721 (0.1285) | **0.0534 (0.0041)** | 0.3229 (0.0087) | 0.1471 (0.0059) |
| $N(0.5 * 1_p, \Sigma)$ | 50,000 | 100 | .2 | 0.1189 (0.0195) | **0.1148 (0.0234)** | 0.3241 (0.0088) | 0.1426 (0.0113) |
| Cauchy$(0.5 * 1_p)$ | 50,000 | 100 | .2 | 0.0738 (0.0053) | **0.0525 (0.0029)** | 0.1045 (0.0071) | 0.0633 (0.0042) |

Table 1 shows the performances of JS-GAN, TV-GAN, dimension halving, and iterative filtering. The network structure, for both JS-GAN and TV-GAN, has one hidden layer with 20 hidden units when the sample size is 50,000 and 2 hidden units when sample size is 5,000. The critical hyper-parameters we apply is given in Appendix and it turns out that the choice of the hyper-parameter is robust against different models when the net structures are the same. To summarize, our method outperforms other algorithms in most cases. TV-GAN is good at cases when $Q$ and $N(0_p, I_p)$ are non-separable but fails when $Q$ is far away from $N(0_p, I_p)$ due to optimization issues discussed in Section 3.1 (Figure 1). On the other hand, JS-GAN stably achieves the lowest error in separable cases and also shows competitive performances for non-separable ones.

## 5.4 NETWORK STRUCTURES

We further study the performance of JS-GAN with various structures of neural networks. The main observation is tuning networks with one-hidden layer becomes tough as the dimension grows (e.g. $p \geq 200$), while a deeper network can significantly refine the situation perhaps by improving the landscape. Some experiment results are given in Table 2. On the other hand, one-hidden layer performs not worse than deeper networks when dimension is not very large (e.g. $p \leq 100$). More experiments are given in Appendix B.4. Additional theoretical results for deep neural nets are given in Appendix A.

Table 2: Experiment results for JS-GAN using networks with different structures in high dimension. Settings: $\epsilon = 0.2$, $p \in \{200, 400\}$ and $n = 50,000$.

| $p$ | 200-100-20-1 | 200-200-100-1 | 200-100-1 | 200-20-1 |
|---|---|---|---|---|
| 200 | 0.0910 (0.0056) | **0.0790 (0.0026)** | 0.3064 (0.0077) | 0.1573 (0.0815) |

| $p$ | 400-200-100-50-20-1 | 400-200-100-20-1 | 400-200-20-1 | 400-200-1 |
|---|---|---|---|---|
| 400 | 0.1477 (0.0053) | 0.1732 (0.0397) | **0.1393 (0.0090)** | 0.3604 (0.0990) |

## 5.5 ADAPTATION TO UNKNOWN COVARIANCE

The robust mean estimator constructed through JS-GAN can be easily made adaptive to unknown covariance structure, which is a special case of (16). We define

$$(\widehat{\theta}, \widehat{\Sigma}) = \operatorname*{argmin}_{\eta \in \mathbb{R}^p, \Gamma \in \mathcal{E}_p} \max_{D \in \mathcal{D}} \left[ \frac{1}{n} \sum_{i=1}^{n} \log D(X_i) + E_{N(\eta,\Gamma)} \log(1 - D(X_i)) \right] + \log 4,$$

The estimator $\widehat{\theta}$, as a result, is rate-optimal even when the true covariance matrix is not necessarily identity and is unknown (see Theorem 4.1). Below, we demonstrate some numerical evidence of the optimality of $\widehat{\theta}$ as well as the error of $\widehat{\Sigma}$ in Table 3.

| Data generating process | Network structure | $\|\widehat{\theta} - 0_p\|$ | $\|\widehat{\Sigma} - \Sigma_1\|_{\mathrm{op}}$ |
|---|---|---|---|
| $0.8N(0_p, \Sigma_1) + 0.2N(0.5 * 1_p, \Sigma_2)$ | 100-20-1 | 0.1680 (0.1540) | 1.9716 (0.7405) |
| $0.8N(0_p, \Sigma_1) + 0.2N(0.5 * 1_p, \Sigma_2)$ | 100-20-20-1 | 0.1824 (0.3034) | 1.4495 (0.6028) |
| $0.8N(0_p, \Sigma_1) + 0.2N(1_p, \Sigma_2)$ | 100-20-1 | 0.0817 (0.0213) | 1.2753 (0.4523) |
| $0.8N(0_p, \Sigma_1) + 0.2N(6 * 1_p, \Sigma_2)$ | 100-20-1 | 0.1069 (0.0357) | 1.1668 (0.1839) |
| $0.8N(0_p, \Sigma_1) + 0.2\mathrm{Cauchy}(0.5 * 1_p)$ | 100-20-1 | 0.0797 (0.0257) | 4.0653 (0.1569) |

Table 3: Numerical experiments for robust mean estimation with unknown covariance trained with $50,000$ samples. The covariance matrices $\Sigma_1$ and $\Sigma_2$ are generated by the same way described in Appendix B.2.

## 5.6 ADAPTATION TO ELLIPTICAL DISTRIBUTIONS

We consider the estimation of the location parameter $\theta$ in elliptical distribution $EC(\theta, \Sigma, h)$ by the JS-GAN defined in (16). In particular, we study the case with i.i.d. observations $X_1, ..., X_n \sim (1-\epsilon)\mathrm{Cauchy}(\theta, I_p) + \epsilon Q$. The density function of $\mathrm{Cauchy}(\theta, \Sigma)$ is given by $p(x; \theta, \Sigma) \propto |\Sigma|^{-1/2} \left(1 + (x - \theta)^T \Sigma^{-1} (x - \theta)\right)^{-(1+p)/2}$.

Compared with Algorithm (1), the difference lies in the choice of the generator. We consider the generator $G_1(\xi, U) = g_\omega(\xi)U + \theta$, where $g_\omega(\xi)$ is a non-negative neural network parametrized by $\omega$ and some random variable $\xi$. The random vector $U$ is sampled from the uniform distribution on $\{u \in \mathbb{R}^p : \|u\| = 1\}$. If the scatter matrix is unknown, we will use the generator $G_2(\xi, U) = g_\omega(\xi)AU + \theta$, with $AA^T$ modeling the scatter matrix.

Table 4 shows the comparison with other methods. Our method still works well under Cauchy distribution, while the performance of other methods that rely on moment conditions deteriorates in this setting.

Table 4: Comparison of various methods of robust location estimation under Cauchy distributions. Samples are drawn from $(1 - \epsilon)\mathrm{Cauchy}(0_p, I_p) + \epsilon Q$ with $\epsilon = 0.2, p = 50$ and various choices of $Q$. Sample size: 50,000. Discriminator net structure: 50-50-25-1. Generator $g_\omega(\xi)$ structure: 48-48-32-24-12-1 with absolute value activation function in the output layer.

| Contamination $Q$ | JS-GAN ($G_1$) | JS-GAN ($G_2$) | Dimension Halving | Iterative Filtering |
|---|---|---|---|---|
| $\mathrm{Cauchy}(1.5 * 1_p, I_p)$ | **0.0664 (0.0065)** | 0.0743 (0.0103) | 0.3529 (0.0543) | 0.1244 (0.0114) |
| $\mathrm{Cauchy}(5.0 * 1_p, I_p)$ | **0.0480 (0.0058)** | 0.0540 (0.0064) | 0.4855 (0.0616) | 0.1687 (0.0310) |
| $\mathrm{Cauchy}(1.5 * 1_p, 5 * I_p)$ | 0.0754 (0.0135) | **0.0742 (0.0111)** | 0.3726 (0.0530) | 0.1220 (0.0112) |
| $\mathrm{Normal}(1.5 * 1_p, 5 * I_p)$ | **0.0702 (0.0064)** | 0.0713 (0.0088) | 0.3915 (0.0232) | 0.1048 (0.0288)) |

## ACKNOWLEDGEMENT

The research of Chao Gao was supported in part by NSF grant DMS-1712957 and NSF Career Award DMS-1847590. The research of Yuan Yao was supported in part by Hong Kong Research Grant Council (HKRGC) grant 16303817, National Basic Research Program of China (No. 2015CB85600), National Natural Science Foundation of China (No. 61370004, 11421110001), as well as awards from Tencent AI Lab, Si Family Foundation, Baidu Big Data Institute, and Microsoft Research-Asia.

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

## A    ADDITIONAL THEORETICAL RESULTS

In this section, we investigate the performance of discriminator classes of deep neural nets with the ReLU activation function. Since our goal is to learn a $p$-dimensional mean vector, a deep neural network discriminator without any regularization will certainly lead to overfitting. Therefore, it is crucial to design a network class with some appropriate regularizations. Inspired by the work of Bartlett (1997); Bartlett & Mendelson (2002), we consider a network class with $\ell_1$ regularizations on all layers except for the second last layer with an $\ell_2$ regularization. With $\mathcal{G}_1^H(B) = \{g(x) = \mathsf{ReLU}(v^T x) : \|v\|_1 \leq B\}$, a neural network class with $l + 1$ layers is defined as

$$\mathcal{G}_{l+1}^H(B) = \left\{ g(x) = \mathsf{ReLU}\left( \sum_{h=1}^{H} v_h g_h(x) \right) : \sum_{h=1}^{H} |v_h| \leq B, g_h \in \mathcal{G}_l^H(B) \right\}.$$

Combining with the last sigmoid layer, we obtain the following discriminator class,

$$\mathcal{F}_L^H(\kappa, \tau, B) = \left\{ D(x) = \mathsf{sigmoid}\left( \sum_{j \geq 1} w_j \mathsf{sigmoid}\left( \sum_{h=1}^{2p} u_{jh} g_{jh}(x) + b_j \right) \right) : \right.$$
$$\left. \sum_{j \geq 1} |w_j| \leq \kappa, \sum_{h=1}^{2p} u_{jh}^2 \leq 2, |b_j| \leq \tau, g_{jh} \in \mathcal{G}_{L-1}^H(B) \right\}.$$

Note that all the activation functions are $\mathsf{ReLU}(\cdot)$ except that we use $\mathsf{sigmoid}(\cdot)$ in the last layer in the feature map $g(\cdot)$. A theoretical guarantees of the class defined above is given by the following theorem.

**Theorem A.1.** *Assume $\frac{p \log p}{n} \vee \epsilon^2 \leq c$ for some sufficiently small constant $c > 0$. Consider i.i.d. observations $X_1, ..., X_n \sim (1 - \epsilon) N(\theta, I_p) + \epsilon Q$ and the estimator $\widehat{\theta}$ defined by (14) with $\mathcal{D} = \mathcal{F}_L^H(\kappa, \tau, B)$ with $H \geq 2p$, $2 \leq L = O(1)$, $2 \leq B = O(1)$, and $\tau = \sqrt{p \log p}$. We set $\kappa = O\left( \sqrt{\frac{p \log p}{n}} + \epsilon \right)$. Then, for the estimator $\widehat{\theta}$ defined by (14) with $\mathcal{D} = \mathcal{F}_L^H(\kappa, \tau, B)$, we have*

$$\|\widehat{\theta} - \theta\|^2 \leq C\left( \frac{p \log p}{n} \vee \epsilon^2 \right),$$

*with probability at least $1 - e^{-C'(p \log p + n\epsilon^2)}$ uniformly over all $\theta \in \mathbb{R}^p$ such that $\|\theta\|_\infty \leq \sqrt{\log p}$ and all $Q$.*

The theorem shows that JS-GAN with a deep ReLU network can achieve the error rate $\frac{p \log p}{n} \vee \epsilon^2$ with respect to the squared $\ell_2$ loss. The condition $\|\theta\|_\infty \leq \sqrt{\log p}$ for the ReLU network can be easily satisfied with a simple preprocessing step. We split the data into two halves, whose sizes are $\log n$ and $n - \log n$, respectively. Then, we calculate the coordinatewise median $\widetilde{\theta}$ using the small half. It is easy to show that $\|\widetilde{\theta} - \theta\|_\infty \leq \sqrt{\frac{\log p}{\log n}} \vee \epsilon$ with high probability. Then, for each $X_i$ from the second half, the conditional distribution of $X_i - \widetilde{\theta}$ given the first half is $(1 - \epsilon) N(\theta - \widetilde{\theta}, I_p) + \epsilon \widetilde{Q}$. Since $\sqrt{\frac{\log p}{\log n}} \vee \epsilon \leq \sqrt{\log p}$, the condition $\|\theta - \widetilde{\theta}\|_\infty \leq \sqrt{\log p}$ is satisfied, and thus we can apply the estimator (14) using the shifted data $X_i - \widetilde{\theta}$ from the second half. The theoretical guarantee of Theorem A.1 will be

$$\|\widehat{\theta} - (\theta - \widetilde{\theta})\|^2 \leq C\left( \frac{p \log p}{n} \vee \epsilon^2 \right),$$

with high probability. Hence, we can use $\widehat{\theta} + \widetilde{\theta}$ as the final estimator to achieve the same rate in Theorem A.1.

On the other hand, our experiments show that this preprocessing step is not needed. We believe that the assumption $\|\theta\|_\infty \leq \sqrt{\log p}$ is a technical artifact in the analysis of the Rademacher complexity. It can probably be dropped by a more careful analysis.

# B  DETAILS OF EXPERIMENTS

## B.1  TRAINING DETAILS

The implementation for JS-GAN is given in Algorithm 1, and a simple modification of the objective function leads to that of TV-GAN.

---

**Algorithm 1** JS-GAN: $\operatorname{argmin}_\eta \max_w [\frac{1}{n} \sum_{i=1}^n \log D_w(X_i) + \mathbb{E} \log(1 - D_w(G_\eta(Z)))]$

---

**Input**: Observation set $S = \{X_1, \ldots, X_n\} \in \mathbb{R}^p$, discriminator network $D_w(x)$, generator network $G_\eta(z) = z + \eta$, learning rates $\gamma_d$ and $\gamma_g$ for the discriminator and the generator, batch size $m$, discriminator steps in each iteration $K$, total epochs $T$, average epochs $T_0$.

**Initialization**: Initialize $\eta$ with coordinatewise median of $S$. Initialize $w$ with $N(0, .05)$ independently on each element or Xavier (Glorot & Bengio, 2010).

1: **for** $t = 1, \ldots, T$ **do**
2:     **for** $k = 1, \ldots, K$ **do**
3:         Sample mini-batch $\{X_1, \ldots, X_m\}$ from $S$. Sample $\{Z_1, \ldots, Z_m\}$ from $N(0, I_p)$
4:         $g_w \leftarrow \nabla_w [\frac{1}{m} \Sigma_{i=1}^m \log D_w(X_i) + \frac{1}{m} \Sigma_{i=1}^m \log(1 - D_w(G_\eta(Z_i)))]$
5:         $w \leftarrow w + \gamma_d g_w$
6:     **end for**
7:     Sample $\{Z_1, \ldots, Z_m\}$ from $N(0, I_p)$
8:     $g_\eta \leftarrow \nabla_\eta [\frac{1}{m} \Sigma_{i=1}^m \log(1 - D_w(G_\eta(Z_i)))]$
9:     $\eta \leftarrow \eta - \gamma_g g_\eta$
10: **end for**

**Return**: The average estimate $\eta$ over the last $T_0$ epochs.

---

Several important implementation details are discussed below.

- *How to tune parameters?* The choice of learning rates is crucial to the convergence rate, but the minimax game is hard to evaluate. We propose a simple strategy to tune hyper-parameters including the learning rates. Suppose we have estimators $\widehat{\theta}_1, \ldots, \widehat{\theta}_M$ with corresponding discriminator networks $D_{\widehat{w}_1}, \ldots, D_{\widehat{w}_M}$. Fixing $\eta = \widehat{\theta}$, we further apply gradient descent to $D_w$ with a few more epochs (but not many in order to prevent overfitting, for example 10 epochs) and select the $\widehat{\theta}$ with the smallest value of the objective function (14) (JS-GAN) or (12) (TV-GAN). We note that training discriminator and generator alternatively usually will not suffer from overfitting since the objective function for either the discriminator or the generator is always changing. However, we must be careful about the overfitting issue when training the discriminator alone with a fixed $\eta$, and that is why we apply an early stopping strategy here. Fortunately, the experiments show if the structures of networks are same (then of course, the dimensions of the inputs are same), the choices of hyper-parameters are robust to different models and we present the critical parameters in Table 5 to reproduce the experiment results in Table 1 and Table 2.

- *When to stop training?* Judging convergence is a difficult task in GAN trainings, since sometimes oscillation may occur. In computer vision, people often use a task related measure and stop training once the requirement based on the measure is achieved. In our experiments below, we simply use a sufficiently large $T$ which works well, but it is still interesting to explore an efficient early stopping rule in the future work.

- *How to design the network structure?* Although Theorem 3.1 and Theorem 3.2 guarantee the minimax rates of TV-GAN without hidden layer and JS-GAN with one hidden layer, one may wonder whether deeper network structures will perform better. From our preliminary experiments, TV-GAN with one hidden layer is significantly better than TV-GAN without any hidden layer. Moreover, JS-GAN

Table 5: Choices of hyper-parameters. The parameter $\lambda$ is the penalty factor for the regularization term (17) and other parameters are listed in Algorithm 1. We apply Xavier initialization (Glorot & Bengio, 2010) for both JS-GAN and TV-GAN trainings.

| Structure | Net | $\gamma_g$ | $\gamma_d$ | $K$ | $T$ | $T_0$ | $\lambda$ |
|---|---|---|---|---|---|---|---|
| 100-20-1 | JS | 0.02 | 0.2 | 5 | 150 | 25 | 0 |
| | TV | 0.0001 | 0.3 | 2 | 150 | 1 | 0.1 |
| 100-2-1 | JS | 0.01 | 0.2 | 5 | 150 | 25 | 0 |
| | TV | 0.01 | 0.1 | 5 | 150 | 1 | 1 |
| 200-20-1 | JS | 0.02 | 0.2 | 5 | 200 | 25 | 0 |
| | TV | 0.0001 | 0.1 | 2 | 200 | 1 | 0.5 |
| 200-200-100-1 | JS | 0.005 | 0.1 | 2 | 200 | 25 | 0 |
| 400-200-20-1 | JS | 0.02 | 0.05 | 2 | 250 | 25 | 0.5 |

with deep network structures can significantly improve over shallow networks especially when the dimension is large (e.g. $p \geq 200$). For a network with one hidden layer, the choice of width may depend on the sample size. If we only have 5,000 samples of 100 dimensions, two hidden units performs better than five hidden units, which performs better than twenty hidden units. If we have 50,000 samples, networks with twenty hidden units perform the best.

- *How to stabilize and accelerate TV-GAN?* As we have discussed in Section 3.1, TV-GAN has a bad landscape when $N(\theta, I_p)$ and the contamination distribution $Q$ are linearly separable (see Figure 1). An outlier removal step before training TV-GAN may be helpful. Besides, spectral normalization (Miyato et al., 2018) is also worth trying since it can prevent the weight from going to infinity and thus can increase the chance to escape from bad saddle points. To accelerate the optimization of TV-GAN, in all the numerical experiments below, we adopt a regularized version of TV-GAN inspired by Proposition 3.1. Since a good feature extractor should match nonlinear moments of $P = (1 - \epsilon)N(\theta, I_p) + \epsilon Q$ and $N(\eta, I_p)$, we use an additional regularization term that can accelerate training and sometimes even leads to better performances. Specifically, let $D(x) = \mathsf{sigmoid}(w^T \Phi(x))$ be the discriminator network with $w$ being the weights of the output layer and $\Phi_D(x)$ be the corresponding network after removing the output layer from $D(x)$. The quantity $\Phi_D(x)$ is usually viewed as a feature extractor, which naturally leads to the following regularization term (Salimans et al., 2016; Mroueh et al., 2017), defined as

$$r(D, \eta) = \left\| \frac{1}{n} \sum_{i=1}^{n} T(\Phi_D, X_i) - T(\Phi_D, N(\eta, I_p)) \right\|^2, \tag{17}$$

where $T(\Phi, P) = \mathbb{E}_P \Phi(X)$ (moment matching) or $T(\Phi, P) = \mathrm{Median}_{X \sim P} \Phi_D(X)$ (median matching).

## B.2 SETTINGS OF CONTAMINATION $Q$

We introduce the contamination distributions $Q$ used in the experiments. We first consider $Q = N(\mu, I_p)$ with $\mu$ ranges in $\{0.2, 0.5, 1, 5\}$. Note that the total variation distance between $N(0_p, I_p)$ and $N(\mu, I_p)$ is of order $\|0_p - \mu\| = \|\mu\|$. We hope to use different levels of $\|\mu\|$ to test the algorithm and verify the error rate in the worst case. Second, we consider $Q = N(1.5 * 1_p, \Sigma)$ to be a Gaussian distribution with a non-trivial covariance matrix $\Sigma$. The covariance matrix is generated according to the following steps. First generate a sparse precision matrix $\Gamma = (\gamma_{ij})$ with each entry $\gamma_{ij} = z_{ij} * \tau_{ij}, i \leq j$, where $z_{ij}$ and $\tau_{ij}$ are independently generated from Uniform$(0.4, 0.8)$ and Bernoulli$(0.1)$. We then define $\gamma_{ij} = \gamma_{ji}$ for all $i > j$ and $\bar{\Gamma} = \Gamma + (|\min \mathrm{eig}(\Gamma)| + 0.05)I_p$ to make the precision matrix symmetric and positive definite, where $\min \mathrm{eig}(\Gamma)$ is the smallest eigenvalue of $\Gamma$. The covariance matrix is $\Sigma = \bar{\Gamma}^{-1}$. Finally, we consider $Q$ to be a Cauchy

distribution with independent component, and the $j$th component takes a standard Cauchy distribution with location parameter $\tau_j = 0.5$.

## B.3 COMPARISON DETAILS

In Section 5.3, we compare GANs with the *dimension halving* (Lai et al., 2016) and *iterative filtering* (Diakonikolas et al., 2017).

- *Dimension Halving.* Experiments conducted are based on the code from `https://github.com/kal2000/AgnosticMeanAndCovarianceCode`. The only hyper-parameter is the threshold in the outlier removal step, and we take $C = 2$ as suggested in the file outRemSperical.m.
- *Iterative Filtering.* Experiments conducted are based on the code from `https://github.com/hoonose/robust-filter`. We assume $\epsilon$ is known and take other hyper-parameters as suggested in the file filterGaussianMean.m.

## B.4 SUPPLEMENTARY EXPERIMENTS FOR NETWORK STRUCTURES

The experiments are conducted with i.i.d. observations drawn from $(1 - \epsilon)N(0_p, I_p) + \epsilon N(0.5 * 1_p, I_p)$ with $\epsilon = 0.2$. Table 6 summarizes results for $p = 100$, $n \in \{5000, 50000\}$ and various network structures. We observe that TV-GAN that uses neural nets with one hidden layer improves over the performance of that without any hidden layer. This indicates that the landscape of TV-GAN might be improved by a more complicated network structure. However, adding one more layer does not improve the results. For JS-GAN, we omit the results without hidden layer because of its lack of robustness (Proposition 3.1). Deeper networks sometimes improve over shallow networks, but this is not always true. We also observe that the optimal choice of the width of the hidden layer depends on the sample size.

Table 6: Experiment results for JS-GAN and TV-GAN with various network structures.

| Structure | $n$ | JS-GAN | TV-GAN |
|---|---|---|---|
| 100-1 | 50,000 | - | 0.1173 (0.0056) |
| 100-20-1 | 50,000 | 0.0953 (0.0064) | 0.1144 (0.0154) |
| 100-50-1 | 50,000 | 0.2409 (0.0500) | 0.1597 (0.0219) |
| 100-20-20-1 | 50,000 | 0.1131 (0.0855) | 0.1724 (0.0295) |
| 100-1 | 5,000 | - | 0.9818 (0.0417) |
| 100-2-1 | 5,000 | 0.1941 (0.0173) | 0.1941 (0.0173) |
| 100-5-1 | 5,000 | 0.2148 (0.0241) | 0.2244 (0.0238) |
| 100-20-1 | 5,000 | 0.3379 (0.0273) | 0.3336 (0.0186) |

## B.5 TABLES FOR TESTING THE MINIMAX RATES

Tables 7, 8 and 9 show numerical results corresponding to Figure 2.

Table 7: Scenario I: $\sqrt{p/n} < \epsilon$. Setting: $p = 100, n = 50,000$, and $\epsilon$ from 0.05 to 0.20. Network structure of JS-GAN: one hidden layer with 5 hidden units. Network structure of TV-GAN: zero-hidden layer. The number in each cell is the average of $\ell_2$ error $\|\hat{\theta} - \theta\|$ with standard deviation in parenthesis estimated from 10 repeated experiments. The bold character marks the worst case among our choices of $Q$ at each $\epsilon$ level. The results of TV-GAN for $Q = N(5 * 1_p, I_p)$ are highlighted in slanted font. The failure of training in this case is due to the bad landscape when $N(0_p, I_p)$ and $Q$ are linearly separable, as discussed in Section 3.1 (see Figure 1).

| $Q$ | Net | $\epsilon = 0.05$ | $\epsilon = 0.10$ | $\epsilon = 0.15$ | $\epsilon = 0.20$ |
|---|---|---|---|---|---|
| $N(0.2 * 1_p, I_p)$ | JS | 0.1025 (0.0080) | 0.1813 (0.0122) | **0.2632 (0.0080)** | **0.3280 (0.0069)** |
| | TV | 0.1110 (0.0204) | 0.2047 (0.0112) | 0.2769 (0.0315) | 0.3283 (0.0745) |
| $N(0.5 * 1_p, I_p)$ | JS | 0.1407 (0.0061) | 0.1895 (0.0070) | 0.1714 (0.0502) | 0.1227 (0.0249) |
| | TV | 0.2003 (0.0480) | 0.2065 (0.1495) | 0.2088 (0.0100) | 0.3985 (0.0112) |
| $N(1_p, I_p)$ | JS | 0.0855 (0.0054) | 0.1055 (0.0322) | 0.0602 (0.0133) | 0.0577 (0.0029) |
| | TV | 0.1084 (0.0063) | 0.0842 (0.0036) | 0.3228 (0.0123) | 0.1329 (0.0125) |
| $N(5 * 1_p, I_p)$ | JS | 0.0587 (0.0033) | 0.0636 (0.0025) | 0.0625 (0.0045) | 0.0591 (0.0040) |
| | TV | *1.2886 (0.5292)* | *4.4511 (0.8754)* | *7.3868 (0.8081)* | *10.5724 (1.2605)* |
| Cauchy$(0.5 * 1_p)$ | JS | 0.0625 (0.0045) | 0.0652 (0.0044) | 0.0648 (0.0035) | 0.0687 (0.0042) |
| | TV | 0.2280 (0.0067) | 0.3842 (0.0083) | 0.5740 (0.0071) | **0.7768 (0.0074)** |
| $N(0.5 * 1_p, \Sigma)$ | JS | **0.1490 (0.0061)** | **0.1958 (0.0074)** | 0.2379 (0.0076) | 0.1973 (0.0679) |
| | TV | **0.2597 (0.0090)** | **0.4621 (0.0649)** | **0.6344 (0.0905)** | 0.7444 (0.3115) |

Table 8: Scenario II-a: $\sqrt{p/n} > \epsilon$. Setting: $n = 1,000, \epsilon = 0.1$, and $p$ from 10 to 100. Other details are the same as above.

| $Q$ | Net | $p = 10$ | $p = 25$ | $p = 50$ | $p = 75$ | $p = 100$ |
|---|---|---|---|---|---|---|
| $N(0.2 * 1_p, I_p)$ | JS | 0.1078 (0.0338) | 0.1819 (0.0215) | 0.3355 (0.0470) | 0.4806 (0.0497) | 0.5310 (0.0414) |
| | TV | 0.2828 (0.0580) | 0.4740 (0.1181) | 0.5627 (0.0894) | 0.8217 (0.0382) | 0.8090 (0.0457) |
| $N(0.5 * 1_p, I_p)$ | JS | 0.1587 (0.0438) | 0.2684 (0.0386) | **0.4213 (0.0356)** | **0.5355 (0.0634)** | 0.6825 (0.0981) |
| | TV | 0.2864 (0.0521) | 0.5024 (0.1038) | 0.6878 (0.1146) | 0.9204 (0.0589) | 0.9418 (0.0551) |
| $N(1_p, I_p)$ | JS | 0.1644 (0.0255) | 0.2177 (0.0480) | 0.3505 (0.0552) | 0.4740 (0.0742) | 0.6662 (0.0611) |
| | TV | **0.3733 (0.0878)** | **0.5407 (0.0634)** | **0.9061 (0.1029)** | **1.0672 (0.0629)** | **1.1150 (0.0942)** |
| $N(5 * 1_p, I_p)$ | JS | 0.0938 (0.0195) | 0.2058 (0.0218) | 0.3316 (0.0462) | 0.4054 (0.0690) | 0.5553 (0.0518) |
| | TV | 0.3707 (0.2102) | 0.7434 (0.3313) | 1.1532 (0.3488) | 1.1850 (0.3739) | 1.3257 (0.1721) |
| Cauchy$(0.5 * 1_p)$ | JS | 0.1188 (0.0263) | 0.1855 (0.0282) | 0.2967 (0.0284) | 0.4094 (0.0385) | 0.4826 (0.0479) |
| | TV | 0.3198 (0.1543) | 0.5205 (0.1049) | 0.6240 (0.0652) | 0.7536 (0.0673) | 0.7612 (0.0613) |
| $N(0.5 * 1_p, \Sigma)$ | JS | **0.1805 (0.0220)** | **0.2692 (0.0318)** | 0.3885 (0.0339) | 0.5144 (0.0547) | **0.6833 (0.1094)** |
| | TV | 0.3036 (0.0736) | 0.5152 (0.0707) | 0.7305 (0.0966) | 0.9460 (0.0900) | 1.0888 (0.0863) |

Table 9: Scenario II-b: $\sqrt{p/n} > \epsilon$. Setting: $p = 50, \epsilon = 0.1$, and $n$ from 50 to $1,000$. Other details are the same as above.

| $Q$ | Net | $n = 50$ | $n = 100$ | $n = 200$ | $n = 500$ | $n = 1000$ |
|---|---|---|---|---|---|---|
| $N(0.2 * 1_p, I_p)$ | JS | 1.3934 (0.5692) | 1.0055 (0.1040) | 0.8373 (0.1335) | 0.4781 (0.0677) | 0.3213 (0.0401) |
| | TV | 1.9714 (0.1552) | 1.2629 (0.0882) | 0.7579 (0.0486) | 0.6640 (0.0689) | 0.6348 (0.0547) |
| $N(0.5 * 1_p, I_p)$ | JS | 1.6422 (0.6822) | 1.2101 (0.2826) | 0.8374 (0.1021) | **0.5832 (0.0595)** | 0.3930 (0.0485) |
| | TV | 1.9780 (0.2157) | 1.2485 (0.0668) | 0.8198 (0.0778) | 0.7597 (0.0456) | 0.7346 (0.0750) |
| $N(1_p, I_p)$ | JS | 1.8427 (0.9633) | 1.2179 (0.2782) | **1.0147 (0.2170)** | 0.5586 (0.1013) | 0.3639 (0.0464) |
| | TV | 1.9907 (0.1498) | **1.4575 (0.1270)** | **0.9724 (0.0802)** | **0.9050 (0.1479)** | **0.8747 (0.0757)** |
| $N(5 * 1_p, I_p)$ | JS | **2.6392 (1.3877)** | **1.3966 (0.5370)** | 0.9633 (0.1383) | 0.5360 (0.0808) | 0.3265 (0.0336) |
| | TV | 2.1050 (0.3763) | 1.5205 (0.2221) | 1.1909 (0.2273) | 1.0957 (0.1390) | 1.0695 (0.2639) |
| Cauchy$(0.5 * 1_p)$ | JS | 1.6563 (0.5246) | 1.0857 (0.3613) | 0.8944 (0.1759) | 0.5363 (0.0593) | 0.3832 (0.0408) |
| | TV | **2.1031 (0.2300)** | 1.1712 (0.1493) | 0.6904 (0.0763) | 0.6300 (0.0642) | 0.5085 (0.0662) |
| $N(0.5 * 1_p, \Sigma)$ | JS | 1.2296 (0.3157) | 0.7696 (0.0786) | 0.5892 (0.0931) | 0.5015 (0.0831) | **0.4085 (0.0209)** |
| | TV | 1.9243 (0.2079) | 1.2217 (0.0681) | 0.7939 (0.0688) | 0.7033 (0.0414) | 0.7125 (0.0490) |

## C    PROOFS OF PROPOSITION 2.1 AND PROPOSITION 3.1

In the first example, consider

$$\mathcal{Q} = \{N(\eta, I_p) : \eta \in \mathbb{R}^p\}, \quad \widetilde{\mathcal{Q}}_\eta = \{N(\widetilde{\eta}, I_p) : \|\widetilde{\eta} - \eta\| \leq r\}.$$

In other words, $\mathcal{Q}$ is the class of Gaussian location family, and $\widetilde{\mathcal{Q}}_\eta$ is taken to be a subset in a local neighborhood of $N(\eta, I_p)$. Then, with $Q = N(\eta, I_p)$ and $\widetilde{Q} = N(\widetilde{\eta}, I_p)$, the event $\widetilde{q}(X)/q(X) \geq 1$ is equivalent to $\|X - \widetilde{\eta}\|^2 \leq \|X - \eta\|^2$. Since $\|\widetilde{\eta} - \eta\| \leq r$, we can write $\widetilde{\eta} = \eta + \widetilde{r}u$ for some $\widetilde{r} \in \mathbb{R}$ and $u \in \mathbb{R}^p$ that satisfy $0 \leq \widetilde{r} \leq r$ and $\|u\| = 1$. Then, (8) becomes

$$\widehat{\theta} = \operatorname*{argmin}_{\eta \in \mathbb{R}^p} \sup_{\substack{\|u\|=1 \\ 0 \leq \widetilde{r} \leq r}} \left[ \frac{1}{n} \sum_{i=1}^n \mathbb{I}\left\{ u^T(X_i - \eta) \geq \frac{\widetilde{r}}{2} \right\} - \mathbb{P}\left( N(0,1) \geq \frac{\widetilde{r}}{2} \right) \right]. \tag{18}$$

Letting $r \to 0$, we obtain (9), the exact formula of Tukey's median.

The next example is a linear model $y|X \sim N(X^T\theta, 1)$. Consider the following classes

$$\begin{aligned} \mathcal{Q} &= \left\{ P_{y,X} = P_{y|X}P_X : P_{y|X} = N(X^T\eta, 1), \eta \in \mathbb{R}^p \right\}, \\ \widetilde{\mathcal{Q}}_\eta &= \left\{ P_{y,X} = P_{y|X}P_X : P_{y|X} = N(X^T\widetilde{\eta}, 1), \|\widetilde{\eta} - \eta\| \leq r \right\}. \end{aligned}$$

Here, $P_{y,X}$ stands for the joint distribution of $y$ and $X$. The two classes $\mathcal{Q}$ and $\widetilde{\mathcal{Q}}$ share the same marginal distribution $P_X$ and the conditional distributions are specified by $N(X^T\eta, 1)$ and $N(X^T\widetilde{\eta}, 1)$, respectively. Follow the same derivation of Tukey's median, let $r \to 0$, and we obtain the exact formula of regression depth (10). It is worth noting that the derivation of (10) does not depend on the marginal distribution $P_X$.

The last example is on covariance/scatter matrix estimation. For this task, we set $\mathcal{Q} = \{N(0, \Gamma) : \Gamma \in \mathcal{E}_p\}$, where $\mathcal{E}_p$ is the class of all $p \times p$ covariance matrices. Inspired by the derivations of Tukey depth and regression depth, it is tempting to choose $\widetilde{\mathcal{Q}}$ in the neighborhood of $N(0, \Gamma)$. However, a native choice would lead to a definition that is not even Fisher consistent. We propose a rank-one neighborhood, given by

$$\widetilde{\mathcal{Q}}_\Gamma = \left\{ N(0, \widetilde{\Gamma}) : \widetilde{\Gamma}^{-1} = \Gamma^{-1} + \widetilde{r}uu^T \in \mathcal{E}_p, |\widetilde{r}| \leq r, \|u\| = 1 \right\}. \tag{19}$$

Then, a direct calculation gives

$$\mathbb{I}\left\{ \frac{dN(0,\widetilde{\Gamma})}{dN(0,\Gamma)}(X) \geq 1 \right\} = \mathbb{I}\left\{ \widetilde{r}|u^TX|^2 \leq \log(1 + \widetilde{r}u^T\Gamma u) \right\}. \tag{20}$$

Since $\lim_{\widetilde{r} \to 0} \frac{\log(1 + \widetilde{r} u^T \Gamma u)}{\widetilde{r} u^T \Gamma u} = 1$, the limiting event of (20) is either $\mathbb{I}\{|u^T X|^2 \leq u^T \Gamma u\}$ or $\mathbb{I}\{|u^T X|^2 \geq u^T \Gamma u\}$, depending on whether $\widetilde{r}$ tends to zero from left or from right. Therefore, with the above $\mathcal{Q}$ and $\widetilde{\mathcal{Q}}_\Gamma$, (8) becomes (11) under the limit $r \to 0$. Even though the definition of (19) is given by a rank-one neighborhood of the inverse covariance matrix, the formula (11) can also be derived with $\widetilde{\Gamma}^{-1} = \Gamma^{-1} + \widetilde{r} u u^T$ in (19) replaced by $\widetilde{\Gamma} = \Gamma + \widetilde{r} u u^T$ by applying the Sherman-Morrison formula. A similar formula to (11) in the literature is given by

$$\widehat{\Sigma} = \operatorname*{argmax}_{\Gamma \in \mathcal{E}_p} \inf_{\|u\|=1} \left[ \frac{1}{n} \sum_{i=1}^n \mathbb{I}\{|u^T X_i|^2 \leq \beta u^T \Gamma u\} \wedge \frac{1}{n} \sum_{i=1}^n \mathbb{I}\{|u^T X_i|^2 \geq \beta u^T \Gamma u\} \right], \tag{21}$$

which is recognized as the maximizer of what is known as the matrix depth function (Zhang, 2002; Chen et al., 2018; Paindaveine & Van Bever, 2017). The $\beta$ in (21) is a scalar defined through the equation $\mathbb{P}(N(0,1) \leq \sqrt{\beta}) = 3/4$. It is proved in Chen et al. (2018) that $\widehat{\Sigma}$ achieves the minimax rate under Huber's $\epsilon$-contamination model. While the formula (11) can be derived from TV-Learning with discriminators in the form of $\mathbb{I}\left\{ \frac{dN(0,\widetilde{\Gamma})}{dN(0,\Gamma)}(X) \geq 1 \right\}$, a special case of (6), the formula (21) can be derived directly from TV-GAN with discriminators in the form of $\mathbb{I}\left\{ \frac{dN(0,\beta\widetilde{\Gamma})}{dN(0,\beta\Gamma)}(X) \geq 1 \right\}$ by following a similar rank-one neighborhood argument. This completes the derivation of Proposition 2.1.

To prove Proposition 3.1, we define $F(w) = E_P \log \mathsf{sigmoid}(w^T g(X)) + E_Q \log(1 - \mathsf{sigmoid}(w^T g(X))) + \log 4$, so that $\mathsf{JS}_g(P, Q) = \max_{w \in \mathcal{W}} F(w)$. The gradient and Hessian of $F(w)$ are given by

$$\nabla F(w) = E_P \frac{e^{-w^T g(X)}}{1 + e^{-w^T g(X)}} g(X) - E_Q \frac{e^{w^T g(X)}}{1 + e^{w^T g(X)}} g(X),$$

$$\nabla^2 F(w) = -E_P \frac{e^{w^T g(X)}}{(1 + e^{w^T g(X)})^2} g(X) g(X)^T - E_Q \frac{e^{-w^T g(X)}}{(1 + e^{-w^T g(X)})^2} g(X) g(X)^T.$$

Therefore, $F(w)$ is concave in $w$, and $\max_{w \in \mathcal{W}} F(w)$ is a convex optimization with a convex $\mathcal{W}$. Suppose $\mathsf{JS}_g(P, Q) = 0$. Then $\max_{w \in \mathcal{W}} F(w) = 0 = F(0)$, which implies $\nabla F(0) = 0$, and thus we have $E_P g(X) = E_Q g(X)$. Now suppose $E_P g(X) = E_Q g(X)$, which is equivalent to $\nabla F(0) = 0$. Therefore, $w = 0$ is a stationary point of a concave function, and we have $\mathsf{JS}_g(P, Q) = \max_{w \in \mathcal{W}} F(w) = F(0) = 0$.

# D  PROOFS OF MAIN RESULTS

In this section, we present proofs of all main theorems in the paper. We first establish some useful lemmas in Section D.1, and the the proofs of main theorems will be given in Section D.2.

## D.1  SOME AUXILIARY LEMMAS

**Lemma D.1.** *Given i.i.d. observations $X_1, ..., X_n \sim \mathbb{P}$ and the function class $\mathcal{D}$ defined in (13), we have for any $\delta > 0$,*

$$\sup_{D \in \mathcal{D}} \left| \frac{1}{n} \sum_{i=1}^n D(X_i) - \mathbb{E}D(X) \right| \leq C \left( \sqrt{\frac{p}{n}} + \sqrt{\frac{\log(1/\delta)}{n}} \right),$$

*with probability at least $1 - \delta$ for some universal constant $C > 0$.*

*Proof.* Let $f(X_1, ..., X_n) = \sup_{D \in \mathcal{D}} \left| \frac{1}{n} \sum_{i=1}^n D(X_i) - \mathbb{E}D(X) \right|$. It is clear that $f(X_1, ..., X_n)$ satisfies the bounded difference condition. By McDiarmid's inequality (McDiarmid, 1989), we have

$$f(X_1, ..., X_n) \leq \mathbb{E}f(X_1, ..., X_n) + \sqrt{\frac{\log(1/\delta)}{2n}},$$

with probability at least $1 - \delta$. Using a standard symmetrization technique (Pollard, 2012), we obtain the following bound that involves Rademacher complexity,

$$\mathbb{E}f(X_1, ..., X_n) \leq 2\mathbb{E} \sup_{D \in \mathcal{D}} \left| \frac{1}{n} \sum_{i=1}^{n} \epsilon_i D(X_i) \right|, \tag{22}$$

where $\epsilon_1, ..., \epsilon_n$ are independent Rademacher random variables. The Rademacher complexity can be bounded by Dudley's integral entropy bound, which gives

$$\mathbb{E} \sup_{D \in \mathcal{D}} \left| \frac{1}{n} \sum_{i=1}^{n} \epsilon_i D(X_i) \right| \lesssim \mathbb{E} \frac{1}{\sqrt{n}} \int_0^2 \sqrt{\log \mathcal{N}(\delta, \mathcal{D}, \| \cdot \|_n)} d\delta,$$

where $\mathcal{N}(\delta, \mathcal{D}, \| \cdot \|_n)$ is the $\delta$-covering number of $\mathcal{D}$ with respect to the empirical $\ell_2$ distance $\|f - g\|_n = \sqrt{\frac{1}{n} \sum_{i=1}^{n} (f(X_i) - g(X_i))^2}$. Since the VC-dimension of $\mathcal{D}$ is $O(p)$, we have $\mathcal{N}(\delta, \mathcal{D}, \| \cdot \|_n) \lesssim p(16e/\delta)^{O(p)}$ (see Theorem 2.6.7 of Van Der Vaart & Wellner (1996)). This leads to the bound $\frac{1}{\sqrt{n}} \int_0^2 \sqrt{\log \mathcal{N}(\delta, \mathcal{D}, \| \cdot \|_n)} d\delta \lesssim \sqrt{\frac{p}{n}}$, which gives the desired result. □

**Lemma D.2.** *Given i.i.d. observations $X_1, ..., X_n \sim \mathbb{P}$, and the function class $\mathcal{D}$ defined in (15), we have for any $\delta > 0$,*

$$\sup_{D \in \mathcal{D}} \left| \frac{1}{n} \sum_{i=1}^{n} \log D(X_i) - \mathbb{E} \log D(X) \right| \leq C\kappa \left( \sqrt{\frac{p}{n}} + \sqrt{\frac{\log(1/\delta)}{n}} \right),$$

*with probability at least $1 - \delta$ for some universal constant $C > 0$.*

*Proof.* Let $f(X_1, ..., X_n) = \sup_{D \in \mathcal{D}} \left| \frac{1}{n} \sum_{i=1}^{n} \log D(X_i) - \mathbb{E} \log D(X) \right|$. Since

$$\sup_{D \in \mathcal{D}} \sup_x |\log(2D(x))| \leq \kappa,$$

we have

$$\sup_{x_1, ..., x_n, x_i'} |f(x_1, ..., x_n) - f(x_1, ..., x_{i-1}, x_i', x_{i+1}, ..., x_n)| \leq \frac{2\kappa}{n}.$$

Therefore, by McDiarmid's inequality (McDiarmid, 1989), we have

$$f(X_1, ..., X_n) \leq \mathbb{E}f(X_1, ..., X_n) + \kappa \sqrt{\frac{2 \log(1/\delta)}{n}}, \tag{23}$$

with probability at least $1 - \delta$. By the same argument of (22), it is sufficient to bound the Rademacher complexity $\mathbb{E} \sup_{D \in \mathcal{D}} \left| \frac{1}{n} \sum_{i=1}^{n} \epsilon_i \log(2D(X_i)) \right|$. Since the function $\psi(x) = \log(2\text{sigmoid}(x))$ has Lipschitz constant 1 and satisfies $\psi(0) = 0$, we have

$$\mathbb{E} \sup_{D \in \mathcal{D}} \left| \frac{1}{n} \sum_{i=1}^{n} \epsilon_i \log(2D(X_i)) \right| \leq 2\mathbb{E} \sup_{\sum_{j \geq 1} |w_j| \leq \kappa, u_j \in \mathbb{R}^p, b_j \in \mathbb{R}} \left| \frac{1}{n} \sum_{i=1}^{n} \epsilon_i \sum_{j \geq 1} w_j \sigma(u_j^T X_i + b_j) \right|,$$

which uses Theorem 12 of Bartlett & Mendelson (2002). By Hölder's inequality, we further have

$$\mathbb{E} \sup_{\sum_{j \geq 1} |w_j| \leq \kappa, u_j \in \mathbb{R}^p, b_j \in \mathbb{R}} \left| \frac{1}{n} \sum_{i=1}^{n} \epsilon_i \sum_{j \geq 1} w_j \sigma(u_j^T X_i + b_j) \right|$$

$$\leq \kappa \mathbb{E} \max_{j \geq 1} \sup_{u_j \in \mathbb{R}^p, b_j \in \mathbb{R}} \left| \frac{1}{n} \sum_{i=1}^{n} \epsilon_i \sigma(u_j^T X_i + b_j) \right|$$

$$= \kappa \mathbb{E} \sup_{u \in \mathbb{R}^p, b \in \mathbb{R}} \left| \frac{1}{n} \sum_{i=1}^{n} \epsilon_i \sigma(u^T X_i + b) \right|.$$

Note that for a monotone function $\sigma : \mathbb{R} \to [0, 1]$, the VC-dimension of the class $\{\sigma(u^T x + b) : u \in \mathbb{R}, b \in \mathbb{R}\}$ is $O(p)$. Therefore, by using the same argument of Dudley's integral entropy bound in the proof Lemma D.1, we have

$$\mathbb{E} \sup_{u \in \mathbb{R}^p, b \in \mathbb{R}} \left| \frac{1}{n} \sum_{i=1}^n \epsilon_i \sigma(u^T X_i + b) \right| \lesssim \sqrt{\frac{p}{n}},$$

which leads to the desired result. $\qquad\square$

**Lemma D.3.** *Given i.i.d. observations $X_1, .., X_n \sim N(\theta, I_p)$ and the function class $\mathcal{F}_L^H(\kappa, \tau, B)$. Assume $\|\theta\|_\infty \le \sqrt{\log p}$ and set $\tau = \sqrt{p \log p}$. We have for any $\delta > 0$,*

$$\sup_{D \in \mathcal{F}_L^H(\kappa, \tau, B)} \left| \frac{1}{n} \sum_{i=1}^n \log D(X_i) - \mathbb{E} \log D(X) \right| \le C\kappa \left( (2B)^{L-1} \sqrt{\frac{p \log p}{n}} + \sqrt{\frac{\log(1/\delta)}{n}} \right),$$

*with probability at least $1 - \delta$ for some universal constants $C > 0$.*

*Proof.* Write $f(X_1, ..., X_n) = \sup_{D \in \mathcal{F}_L^H(\kappa, \tau, B)} \left| \frac{1}{n} \sum_{i=1}^n \log D(X_i) - \mathbb{E} \log D(X) \right|$. Then, the inequality (23) holds with probability at least $1 - \delta$. It is sufficient to analyze the Rademacher complexity. Using the fact that the function $\log(2\mathsf{sigmoid}(x))$ is Lipschitz and Hölder's inequality, we have

$$\mathbb{E} \sup_{D \in \mathcal{F}_L^H(\kappa, \tau, B)} \left| \frac{1}{n} \sum_{i=1}^n \epsilon_i \log(2D(X_i)) \right|$$

$$\le 2\mathbb{E} \sup_{\|w\|_1 \le \kappa, \|u_{j*}\|^2 \le 2, |b_j| \le \tau, g_{jh} \in \mathcal{G}_{L-1}^H(B)} \left| \frac{1}{n} \sum_{i=1}^n \epsilon_i \sum_{j \ge 1} w_j \mathsf{sigmoid} \left( \sum_{h=1}^{2p} u_{jh} g_{jh}(X_i) + b_j \right) \right|$$

$$\le 2\kappa\mathbb{E} \sup_{\|u\|^2 \le 2, |b| \le \tau, g_h \in \mathcal{G}_{L-1}^H(B)} \left| \frac{1}{n} \sum_{i=1}^n \epsilon_i \mathsf{sigmoid} \left( \sum_{h=1}^{2p} u_h g_h(X_i) + b \right) \right|$$

$$\le 4\kappa\mathbb{E} \sup_{\|u\|^2 \le 2, |b| \le \tau, g_h \in \mathcal{G}_{L-1}^H(B)} \left| \frac{1}{n} \sum_{i=1}^n \epsilon_i \left( \sum_{h=1}^{2p} u_h g_h(X_i) + b \right) \right|$$

$$\le 8\sqrt{p}\kappa\mathbb{E} \sup_{g \in \mathcal{G}_{L-1}^H(B)} \left| \frac{1}{n} \sum_{i=1}^n \epsilon_i g(X_i) \right| + 4\kappa\tau\mathbb{E} \left| \frac{1}{n} \sum_{i=1}^n \epsilon_i \right|.$$

Now we use the notation $Z_i = X_i - \theta \sim N(0, I_p)$ for $i = 1, ..., n$. We bound $\mathbb{E} \sup_{g \in \mathcal{G}_{L-1}^H(B)} \left| \frac{1}{n} \sum_{i=1}^n \epsilon_i g(Z_i + \theta) \right|$ by induction. Since

$$\mathbb{E} \left( \sup_{g \in \mathcal{G}_1^H(B)} \frac{1}{n} \sum_{i=1}^n \epsilon_i g(Z_i + \theta) \right)$$

$$\le \mathbb{E} \left( \sup_{\|v\|_1 \le B} \frac{1}{n} \sum_{i=1}^n \epsilon_i v^T(Z_i + \theta) \right)$$

$$\le B \left( \mathbb{E} \left| \frac{1}{n} \sum_{i=1}^n \epsilon_i Z_i \right|_\infty + \|\theta\|_\infty \mathbb{E} \left| \frac{1}{n} \sum_{i=1}^n \epsilon_i \right| \right)$$

$$\le CB \frac{\sqrt{\log p} + \|\theta\|_\infty}{\sqrt{n}},$$

and

$$\mathbb{E}\left(\sup_{g \in \mathcal{G}_{l+1}^H(B)} \frac{1}{n} \sum_{i=1}^n \epsilon_i g(Z_i + \theta)\right)$$

$$\leq \quad \mathbb{E}\left(\sup_{\|v\|_1 \leq B, g_h \in \mathcal{G}_l^H(B)} \frac{1}{n} \sum_{i=1}^n \epsilon_i \sum_{h=1}^H v_h g_h(Z_i + \theta)\right)$$

$$\leq \quad B\mathbb{E}\left(\sup_{g \in \mathcal{G}_l^H(B)} \left|\frac{1}{n} \sum_{i=1}^n \epsilon_i g(Z_i + \theta)\right|\right)$$

$$\leq \quad 2B\mathbb{E}\left(\sup_{g \in \mathcal{G}_l^H(B)} \frac{1}{n} \sum_{i=1}^n \epsilon_i g(Z_i + \theta)\right),$$

we have

$$\mathbb{E}\left(\sup_{g \in \mathcal{G}_{L-1}^H(B)} \frac{1}{n} \sum_{i=1}^n \epsilon_i g(Z_i + \theta)\right) \leq C(2B)^{L-1} \frac{\sqrt{\log p} + \|\theta\|_\infty}{\sqrt{n}}.$$

Combining the above inequalities, we get

$$\mathbb{E}\left(\sup_{D \in \mathcal{F}_L^H(\kappa, \tau, B)} \frac{1}{n} \sum_{i=1}^n \epsilon_i \log D(Z_i + \theta)\right) \leq C\kappa \left(\sqrt{p}(2B)^{L-1} \frac{\sqrt{\log p} + \|\theta\|_\infty}{\sqrt{n}} + \frac{\tau}{\sqrt{n}}\right).$$

This leads to the desired result under the conditions on $\tau$ and $\|\theta\|_\infty$. $\qquad\qquad\square$

## D.2 PROOFS OF MAIN THEOREMS

*Proof of Theorem 3.1.* We first introduce some notations. Define $F(P, \eta) = \max_{w,b} F_{w,b}(P, \eta)$, where

$$F_{w,b}(P, \eta) = E_P \mathsf{sigmoid}(w^T X + b) - E_{N(\eta, I_p)} \mathsf{sigmoid}(w^T X + b).$$

With this definition, we have $\widehat{\theta} = \operatorname{argmin}_\eta F(\mathbb{P}_n, \eta)$, where we use $\mathbb{P}_n$ for the empirical distribution $\frac{1}{n} \sum_{i=1}^n \delta_{X_i}$. We shorthand $N(\eta, I_p)$ by $P_\eta$, and then

$$F(P_\theta, \widehat{\theta}) \leq F((1-\epsilon)P_\theta + \epsilon Q, \widehat{\theta}) + \epsilon \qquad\qquad (24)$$

$$\leq F(\mathbb{P}_n, \widehat{\theta}) + \epsilon + C\left(\sqrt{\frac{p}{n}} + \sqrt{\frac{\log(1/\delta)}{n}}\right) \qquad\qquad (25)$$

$$\leq F(\mathbb{P}_n, \theta) + \epsilon + C\left(\sqrt{\frac{p}{n}} + \sqrt{\frac{\log(1/\delta)}{n}}\right) \qquad\qquad (26)$$

$$\leq F((1-\epsilon)P_\theta + \epsilon Q, \theta) + \epsilon + 2C\left(\sqrt{\frac{p}{n}} + \sqrt{\frac{\log(1/\delta)}{n}}\right) \qquad\qquad (27)$$

$$\leq F(P_\theta, \theta) + 2\epsilon + 2C\left(\sqrt{\frac{p}{n}} + \sqrt{\frac{\log(1/\delta)}{n}}\right) \qquad\qquad (28)$$

$$= 2\epsilon + 2C\left(\sqrt{\frac{p}{n}} + \sqrt{\frac{\log(1/\delta)}{n}}\right). \qquad\qquad (29)$$

With probability at least $1 - \delta$, the above inequalities hold. We will explain each inequality. Since

$$F((1-\epsilon)P_\theta + \epsilon Q, \eta) = \max_{w,b} \left[(1-\epsilon)F_{w,b}(P_\theta, \eta) + \epsilon F_{w,b}(Q, \eta)\right],$$

we have

$$\sup_\eta |F((1-\epsilon)P_\theta + \epsilon Q, \eta) - F(P_\theta, \eta)| \leq \epsilon,$$

which implies (24) and (28). The inequalities (25) and (27) are implied by Lemma D.1 and the fact that

$$\sup_{\eta} |F(\mathbb{P}_n, \eta) - F((1-\epsilon)P_\theta + \epsilon Q, \eta)| \le \sup_{w,b} \left| \frac{1}{n} \sum_{i=1}^{n} \mathsf{sigmoid}(w^T X_i + b) - \mathbb{E}\mathsf{sigmoid}(w^T X + b) \right|.$$

The inequality (26) is a direct consequence of the definition of $\widehat{\theta}$. Finally, it is easy to see that $F(P_\theta, \theta) = 0$, which gives (29). In summary, we have derived that with probability at least $1 - \delta$,

$$F_{w,b}(P_\theta, \widehat{\theta}) \le 2\epsilon + 2C \left( \sqrt{\frac{p}{n}} + \sqrt{\frac{\log(1/\delta)}{n}} \right),$$

for all $w \in \mathbb{R}^p$ and $b \in \mathbb{R}$. For any $u \in \mathbb{R}^p$ such that $\|u\| = 1$, we take $w = u$ and $b = -u^T \theta$, and we have

$$f(0) - f(u^T(\theta - \widehat{\theta})) \le 2\epsilon + 2C \left( \sqrt{\frac{p}{n}} + \sqrt{\frac{\log(1/\delta)}{n}} \right),$$

where $f(t) = \int \frac{1}{1+e^{z+t}} \phi(z) dz$, with $\phi(\cdot)$ being the probability density function of $N(0,1)$. It is not hard to see that as long as $|f(t) - f(0)| \le c$ for some sufficiently small constant $c > 0$, then $|f(t) - f(0)| \ge c'|t|$ for some constant $c' > 0$. This implies

$$
\begin{aligned}
\|\widehat{\theta} - \theta\| &= \sup_{\|u\|=1} |u^T(\widehat{\theta} - \theta)| \\
&\le \frac{1}{c'} \sup_{\|u\|=1} \left| f(0) - f(u^T(\theta - \widehat{\theta})) \right| \\
&\lesssim \epsilon + \sqrt{\frac{p}{n}} + \sqrt{\frac{\log(1/\delta)}{n}},
\end{aligned}
$$

with probability at least $1 - \delta$. The proof is complete. $\qquad\square$

*Proof of Theorem 3.2.* We continue to use $P_\eta$ to denote $N(\eta, I_p)$. Define

$$F(P, \eta) = \max_{\|w\|_1 \le \kappa, u, b} F_{w,u,b}(P, \eta),$$

where

$$F_{w,u,b}(P, \eta) = E_P \log D(X) + E_{N(\eta, I_p)} \log(1 - D(X)) + \log 4,$$

with $D(x) = \mathsf{sigmoid}\left( \sum_{j \ge 1} w_j \sigma(u_j^T x + b_j) \right)$. Then,

$$
\begin{aligned}
F(P_\theta, \widehat{\theta}) &\le F((1-\epsilon)P_\theta + \epsilon Q, \widehat{\theta}) + 2\kappa\epsilon & (30) \\
&\le F(\mathbb{P}_n, \widehat{\theta}) + 2\kappa\epsilon + C\kappa \left( \sqrt{\frac{p}{n}} + \sqrt{\frac{\log(1/\delta)}{n}} \right) & (31) \\
&\le F(\mathbb{P}_n, \theta) + 2\kappa\epsilon + C\kappa \left( \sqrt{\frac{p}{n}} + \sqrt{\frac{\log(1/\delta)}{n}} \right) & (32) \\
&\le F((1-\epsilon)P_\theta + \epsilon Q, \theta) + 2\kappa\epsilon + 2C\kappa \left( \sqrt{\frac{p}{n}} + \sqrt{\frac{\log(1/\delta)}{n}} \right) & (33) \\
&\le F(P_\theta, \theta) + 4\kappa\epsilon + 2C\kappa \left( \sqrt{\frac{p}{n}} + \sqrt{\frac{\log(1/\delta)}{n}} \right) & (34) \\
&= 4\kappa\epsilon + 2C\kappa \left( \sqrt{\frac{p}{n}} + \sqrt{\frac{\log(1/\delta)}{n}} \right).
\end{aligned}
$$

The inequalities (30)-(34) follow similar arguments for (24)-(28). To be specific, (31) and (33) are implied by Lemma D.2, and (32) is a direct consequence of the definition of $\widehat{\theta}$. To see (30) and (34), note that for any $w$ such that $\|w\|_1 \leq \kappa$, we have

$$|\log(2D(X))| \leq \left| \sum_{j \geq 1} w_j \sigma(u_j^T X + b_j) \right| \leq \kappa.$$

A similar argument gives the same bound for $|\log(2(1 - D(X)))|$. This leads to

$$\sup_{\eta} |F((1 - \epsilon)P_\theta + \epsilon Q, \eta) - F(P_\theta, \eta)| \leq 2\kappa\epsilon, \tag{35}$$

which further implies (30) and (34). To summarize, we have derived that with probability at least $1 - \delta$,

$$F_{w,u,b}(P_\theta, \widehat{\theta}) \leq 4\kappa\epsilon + 2C\kappa \left( \sqrt{\frac{p}{n}} + \sqrt{\frac{\log(1/\delta)}{n}} \right),$$

for all $\|w\|_1 \leq \kappa$, $\|u_j\| \leq 1$ and $b_j$. Take $w_1 = \kappa$, $w_j = 0$ for all $j > 1$, $u_1 = u$ for some unit vector $u$ and $b_1 = -u^T\theta$, and we get

$$f_{u^T(\widehat{\theta}-\theta)}(\kappa) \leq 4\kappa\epsilon + 2C\kappa \left( \sqrt{\frac{p}{n}} + \sqrt{\frac{\log(1/\delta)}{n}} \right), \tag{36}$$

where

$$f_\delta(t) = \mathbb{E} \log \frac{2}{1 + e^{-t\sigma(Z)}} + \mathbb{E} \log \frac{2}{1 + e^{t\sigma(Z+\delta)}}, \tag{37}$$

with $Z \sim N(0, 1)$. Direct calculations give

$$
\begin{aligned}
f_\delta'(t) &= \mathbb{E} \frac{e^{-t\sigma(Z)}}{1 + e^{-t\sigma(Z)}} \sigma(Z) - \mathbb{E} \frac{e^{t\sigma(Z+\delta)}}{1 + e^{t\sigma(Z+\delta)}} \sigma(Z + \delta), \\
f_\delta''(t) &= -\mathbb{E}\sigma(Z)^2 \frac{e^{-t\sigma(Z)}}{(1 + e^{-t\sigma(Z)})^2} - \mathbb{E}\sigma(Z + \delta)^2 \frac{e^{t\sigma(Z+\delta)}}{(1 + e^{t\sigma(Z+\delta)})^2}.
\end{aligned}
\tag{38}
$$

Therefore, $f_\delta(0) = 0$, $f_\delta'(0) = \frac{1}{2}(\mathbb{E}\sigma(Z) - \mathbb{E}\sigma(Z + \delta))$, and $f_\delta''(t) \geq -\frac{1}{2}$. By the inequality

$$f_\delta(\kappa) \geq f_\delta(0) + \kappa f_\delta'(0) - \frac{1}{4}\kappa^2,$$

we have $\kappa f_\delta'(0) \leq f_\delta(\kappa) + \kappa^2/4$. In view of (36), we have

$$
\begin{aligned}
&\frac{\kappa}{2} \left( \int \sigma(z)\phi(z)dz - \int \sigma(z + u^T(\widehat{\theta} - \theta))\phi(z)dz \right) \\
&\leq \quad 4\kappa\epsilon + 2C\kappa \left( \sqrt{\frac{p}{n}} + \sqrt{\frac{\log(1/\delta)}{n}} \right) + \frac{\kappa^2}{4}.
\end{aligned}
$$

It is easy to see that for the choices of $\sigma(\cdot)$, $\int \sigma(z)\phi(z)dz - \int \sigma(z + t)\phi(z)dz$ is locally linear with respect to $t$. This implies that

$$\kappa\|\widehat{\theta} - \theta\| = \kappa \sup_{\|u\|=1} u^T(\widehat{\theta} - \theta) \lesssim \kappa \left( \epsilon + \sqrt{\frac{p}{n}} + \sqrt{\frac{\log(1/\delta)}{n}} \right) + \kappa^2.$$

Therefore, with a $\kappa \lesssim \sqrt{\frac{p}{n}} + \epsilon$, the proof is complete. $\qquad\square$

*Proof of Theorem 4.1.* We use $P_{\theta,\Sigma,h}$ to denote the elliptical distribution $EC(\theta, \Sigma, h)$. Define

$$F(P, (\eta, \Gamma, g)) = \max_{\|w\|_1 \leq \kappa, u, b} F_{w,u,b}(P, (\eta, \Gamma, g)),$$

where

$$F_{w,u,b}(P, (\eta, \Gamma, g)) = E_P \log D(X) + E_{EC(\eta, \Gamma, g)} \log (1 - D(X)) + \log 4,$$

with $D(x) = \text{sigmoid}\left(\sum_{j \geq 1} w_j \sigma(u_j^T x + b_j)\right)$. Let $P$ be the data generating process that satisfies $\text{TV}(P, P_{\theta, \Sigma, h}) \leq \epsilon$, and then there exist probability distributions $Q_1$ and $Q_2$, such that

$$P + \epsilon Q_1 = P_{\theta, \Sigma, h} + \epsilon Q_2.$$

The explicit construction of $Q_1, Q_2$ is given in the proof of Theorem 5.1 of Chen et al. (2018). This implies that

$$
\begin{aligned}
&|F(P, (\eta, \Gamma, g)) - F(P_{\theta, \Sigma, h}, (\eta, \Gamma, g))| \\
\leq\quad & \sup_{\|w\|_1 \leq \kappa, u, b} |F_{w,u,b}(P, (\eta, \Gamma, g)) - F_{w,u,b}(P_{\theta, \Sigma, h}, (\eta, \Gamma, g))| \\
=\quad & \epsilon \sup_{\|w\|_1 \leq \kappa, u, b} |E_{Q_2} \log(2D(X)) - E_{Q_1} \log(2D(X))| \\
\leq\quad & 2\kappa\epsilon.
\end{aligned}
\tag{39}
$$

Then, the same argument in Theorem 3.2 (with (35) replaced by (39)) leads to the fact that with probability at least $1 - \delta$,

$$F_{w,u,b}(P_{\theta, \Sigma, h}, (\widehat{\theta}, \widehat{\Sigma}, \widehat{h})) \leq 4\kappa\epsilon + 2C\kappa \left( \sqrt{\frac{p}{n}} + \sqrt{\frac{\log(1/\delta)}{n}} \right),$$

for all $\|w\|_1 \leq \kappa$, $\|u_j\| \leq 1$ and $b_j$. Take $w_1 = \kappa$, $w_j = 0$ for all $j > 1$, $u_1 = u/\sqrt{u^T \widehat{\Sigma} u}$ for some unit vector $u$ and $b_1 = -u^T \theta / \sqrt{u^T \widehat{\Sigma} u}$, and we get

$$f_{\frac{u^T(\widehat{\theta} - \theta)}{\sqrt{u^T \widehat{\Sigma} u}}}(\kappa) \leq 4\kappa\epsilon + 2C\kappa \left( \sqrt{\frac{p}{n}} + \sqrt{\frac{\log(1/\delta)}{n}} \right),$$

where

$$f_\delta(t) = \int \log \left( \frac{2}{1 + e^{-t\sigma(\Delta s)}} \right) h(s) ds + \int \log \left( \frac{2}{1 + e^{t\sigma(\delta + s)}} \right) \widehat{h}(s) ds,$$

where $\delta = \frac{u^T(\widehat{\theta} - \theta)}{\sqrt{u^T \widehat{\Sigma} u}}$ and $\Delta = \frac{\sqrt{u^T \Sigma u}}{\sqrt{u^T \widehat{\Sigma} u}}$. A similar argument to the proof of Theorem 3.2 gives

$$
\begin{aligned}
& \frac{\kappa}{2} \left( \int \sigma(\Delta s) h(s) ds - \int \sigma(\delta + s) \widehat{h}(s) ds \right) \\
\leq\quad & 4\kappa\epsilon + 2C\kappa \left( \sqrt{\frac{p}{n}} + \sqrt{\frac{\log(1/\delta)}{n}} \right) + \frac{\kappa^2}{4}.
\end{aligned}
$$

Since

$$\int \sigma(\Delta s) h(s) ds = \frac{1}{2} = \int \sigma(s) \widehat{h}(s) ds,$$

the above bound is equivalent to

$$\frac{\kappa}{2} (H(0) - H(\delta)) \leq 4\kappa\epsilon + 2C\kappa \left( \sqrt{\frac{p}{n}} + \sqrt{\frac{\log(1/\delta)}{n}} \right) + \frac{\kappa^2}{4},$$

where $H(\delta) = \int \sigma(\delta + s) \widehat{h}(s) ds$. The above bound also holds for $\frac{\kappa}{2}(H(\delta) - H(0))$ by a symmetric argument, and therefore the same bound holds for $\frac{\kappa}{2}|H(\delta) - H(0)|$. Since $H'(0) = \int \sigma(s)(1 - \sigma(s)) \widehat{h}(s) ds = 1$, $H(\delta)$ is locally linear at $\delta = 0$, which leads to a desired bound for $\delta = \frac{u^T(\widehat{\theta} - \theta)}{\sqrt{u^T \widehat{\Sigma} u}}$. Finally, since $u^T \widehat{\Sigma} u \leq M$, we get the bound for $u^T(\widehat{\theta} - \theta)$. The proof is complete by taking supreme over all unit vector $u$. $\square$

*Proof of Theorem A.1.* We continue to use $P_\eta$ to denote $N(\eta, I_p)$. Define

$$F(P, \eta) = \sup_{D \in \mathcal{F}_L^H(\kappa, \tau, B)} F_D(P, \eta),$$

with

$$F_D(P, \eta) = E_P \log D(X) + E_{N(\eta, I_p)} \log(1 - D(X)) + \log 4.$$

Follow the same argument in the proof of Theorem 3.2, use Lemma D.3, and we have

$$F_D(P_\theta, \widehat{\theta}) \le C\kappa \left( \epsilon + (2B)^{L-1} \sqrt{\frac{p \log p}{n}} + \sqrt{\frac{\log(1/\delta)}{n}} \right),$$

uniformly over $D \in \mathcal{F}_L^H(\kappa, \tau, B)$ with probability at least $1 - \delta$. Choose $w_1 = \kappa$ and $w_j = 0$ for all $w_j > 1$. For any unit vector $\widetilde{u} \in \mathbb{R}^p$, take $u_{1h} = -u_{1(h+p)} = \widetilde{u}_h$ for $h = 1, ..., p$ and $b_1 = -\widetilde{u}^T \theta$. For $h = 1, ..., p$, set $g_{1h}(x) = \max(x_h, 0)$. For $h = p+1, ..., 2p$, set $g_{1h}(x) = \max(-x_{h-p}, 0)$. It is obvious that such $u$ and $b$ satisfy $\sum_h u_{1h}^2 \le 2$ and $|b_1| \le \|\theta\| \le \sqrt{p}\|\theta\|_\infty \le \sqrt{p \log p}$. We need to show both the functions $\max(x, 0)$ and $\max(-x, 0)$ are elements of $\mathcal{G}_{L-1}^H(B)$. This can be proved by induction. It is obvious that $\max(x_h, 0), \max(-x_h, 0) \in \mathcal{G}_1^H(B)$ for any $h = 1, ..., p$. Suppose we have $\max(x_h, 0), \max(-x_h, 0) \in \mathcal{G}_l^H(B)$ for any $h = 1, ..., p$. Then,

$$\begin{aligned}
\max\left(\max(x_h, 0) - \max(-x_h, 0), 0\right) &= \max(x_h, 0), \\
\max\left(\max(-x_h, 0) - \max(x_h, 0), 0\right) &= \max(-x_h, 0).
\end{aligned}$$

Therefore, $\max(x_h, 0), \max(-x_h, 0) \in \mathcal{G}_{l+1}^H(B)$ as long as $B \ge 2$. Hence, the above construction satisfies $D(x) = \mathsf{sigmoid}(\kappa \mathsf{sigmoid}(\widetilde{u}^T(x - \theta))) \in \mathcal{F}_L^H(\kappa, \tau, B)$, and we have

$$f_{u^T(\widehat{\theta} - \theta)}(\kappa) \le C\kappa \left( \epsilon + (2B)^{L-1} \sqrt{\frac{p \log p}{n}} + \sqrt{\frac{\log(1/\delta)}{n}} \right), \tag{40}$$

where the definition of $f_\delta(t)$ is given by (37) with $Z \sim N(0, 1)$ and $\sigma(\cdot)$ is taken as $\mathsf{sigmoid}(\cdot)$. Apply the a similar in the proof of Theorem 3.2, we obtain the desired result. $\qquad \square$

