# OpenReview forum: "ROBUST ESTIMATION VIA GENERATIVE ADVERSARIAL NETWORKS"
_ICLR.cc/2019/Conference_

### Official Review · AnonReviewer1 · 2018-11-03
**Interesting paper with important connection between GANs' loss and robust estimation**

**Rating:** 7
**Confidence:** 5

**Review:**

The authors considered Huber contamination model.
They use f-divergence and its variational lower bound to get a criterion for probability distribution function estimation.
They showed that  under different functions f in f-divergence they can get different criteria used in robust depth-based estimation of a mean and/or covariance matrix.
For f, corresponding to the Total Variation divergence and discriminator being a logistic regression, they proved that the robust estimate can achieve the minimax rate, although there could be difficulties to optimize the criterion. Then the authors showed that for the JS-divergence with discriminator in the form of a one-layer neural network we can get robust and optimal estimate, while the criterion itself can be efficiently optimized.

Comments
- it could be good to define what Tau is right after formula (3). Analogously for the class of probability distributions $mathcal{Q}$ in (4), in for $\tilde{\mathcal{Q}}$ in (5)
- page 3, line 12 from above: “and f’(t) = e^{t-1}.” In fact, here we should use $f^*(t)$
- page 3, proposition 2.1, subsection 1 of the proposition: $\tilde{\mathcal{Q}}$ instead of $\tilde{Q}$ should be used as a notation for a class of probability distributions
- in (12) the authors unexpectedly introduced a new notation $D$. I guess they should specify right after formula (12) what is $D$
- theorem 3.1. If it is possible, it could be good at least to speculate on how $C, C’$ depend on $c$ in the displayed formula
- axis labels in figure 2 are almost impossible to read. This somehow should be improved
- in table 1 we clearly see that TV-GAN is better for some part of problems, and JS-GAN is better for another part of problems. Why? Any comments? At least intuition?
- page 8, “ On the other hand, JS-GAN stably achieves the lowest error in separable cases and also shows competitive performances for non-separable ones.” Why? Any comments?

Conclusion
- in general, the paper is well written
- it contains sufficient number of experiments to prove that the proposed approach is reasonable
- the connection between GANs based on f-divergence and robust estimation seems to be important. Thus I’d like to proposed to accept this paper

---

> ### Author Response · Authors · 2018-11-18
> **Response to AnonReviewer1**
>
> Thank you for your comments. The response to each point is listed below:
> - Here, Tau, \mathcal{Q} and \tilde{\mathcal{Q}} are an arbitrary function classes. They will only be specified in specific problems such as mean estimation and covariance matrix estimation.
> - We agree. It should be f^*.
> - We agree. This is a typo.
> - In (12), D(x) is the same as T(x) in (4). The reason we use a new notation is that for JS-GAN, log D(x) in (14) is T(x) in (4). We will make a clarification in the revision.
> - The constants C,C’ do not depend on c anymore, as long as c is smaller than some absolute number, say c<1/100.
> - Theoretically speaking, TV-GAN should be the best, because of its close connection to depth-based estimators. The problem with TV-GAN is its optimization property, which is illustrated discussed in Figure 1. Whenever the contamination distribution is not close to the true model, TV-GAN suffers from this problem, and then it is outperformed by JS-GAN.
> - JS-GAN does not have the optimization difficulty as TV-GAN does. Moreover, we prove that JS-GAN is minimax optimal, and therefore, it has a stable performance and it is the one that we recommend.
>
> The revised manuscript is uploaded with changes highlighted in red.

---

### Official Review · AnonReviewer2 · 2018-11-03
**Interesting, but unclear if deep learning is the right framework for this problem**

**Rating:** 5
**Confidence:** 5

**Review:**

The paper considers the problem of robust high dimensional estimation in Huber’s contamination model. The algorithm is given samples from a distribution (1 - eps) * P + eps * Q, where P is a “nice” distribution (e.g. a Gaussian), eps is the fraction of contaminated points, and Q is some unconstrained noise distribution. The goal is then to estimate parameters of P as well as possible, given this noise. The settings they primarily consider in this paper are when P is a Gaussian with unknown mean and identity covariance, or when it is a Gaussian with unknown covariance. Classical estimators such as Tukey depth or matrix depth for these problems achieve optimal minimax rates, but are computationally expensive to compute. However, recent work of [1,2] propose efficient estimators for this problem that (nearly) achieve these rates.

This paper considers a different approach to this problem. They observe that in the case when P is a Gaussian, these classical depth functions (or minor variations thereof) can be written as the asymptotic limits of certain types of GANs. They then demonstrate that for specific choices for the architecture and regularization of the discriminator, the global optima of this GAN objective achieves minimax optimal error and rates in Huber’s contamination model. Unfortunately, they do not prove that their algorithm achieves these global minima. As a result they do not have any provable guarantees for their algorithms. However, they show experimentally that against many choices of noise distribution, their algorithms obtain good error, both for mean estimation and covariance estimation (at least, the JS-GAN seems to consistently succeed. They acknowledge that the TV-GAN seems to be unstable in certain regimes).

Pros:

- I think the question of finding algorithmic equivalents of Tukey median is a very interesting question, and this is an interesting attempt.
- I did not replicate their experiments on GANs, but the experimental numbers seem promising. However, I have some mixed feelings about this (see below).

Cons:

- A clear disadvantage of the approach to prior algorithmic work is that the algorithms proposed in the paper do not have provable guarantees. For settings such as secure machine learning, the lack of such guarantees is problematic. Given that previous works give efficient (i.e. practical) algorithms for these problems with provable guarantees, I am unclear how much impact this will have in practice.

- Given that TV-GAN is known to fail (as shown in Table 6), it is unclear how useful the numbers for it are in Table 1. Without these numbers, it then appears that JS-GAN and the filtering algorithm often achieve comparable results, although it is very interesting that JS-GAN is consistently slightly better.

- I feel that the authors fall short of their goal to make a good algorithmic analog of these depth-based estimators. This is a subtle but important point, so let me justify this. As the authors explain, the major advantage of such estimators would be that they are model-free: they should give robustness for a number of settings, not just Gaussians, but also elliptical distributions, sub-gaussian distributions, etc. However, the correspondence that the authors derive to their GAN formulation of depth heavily leverages the Gaussianity of the underlying distribution. Specifically, it leverages the fact that the Scheffe set between two Gaussians is a half-plane, which clearly fails for more general distributions. As a result, it appears to me that this variational formulation of depth succeeds only in a very model-specific setting. As a result, from a theoretical perspective it is unclear what advantage this formulation has.


Questions:

- How long does it take to train the GANs? Is it comparable to the runtime of the other algorithms?

- Can these algorithms work in the stronger notions of corruption considered in [1, 2]?

Overall conclusion:

The paper proposes a novel framework for robust estimation. However, in light of the previous provable and much simpler algorithms for robust estimation, in the end it seems to me that deep learning is an unnecessarily complicated approach to this problem. While the authors demonstrate some experimental improvement in the test cases they tried, the lack of provable guarantees for their approach limits the theoretical appeal of their paper. More conceptually, I am unconvinced that their approach is the correct approach to understanding algorithmic notions of depth, for the reasons described above.

[1] Kevin A Lai, Anup B Rao, and Santosh Vempala. Agnostic estimation of mean and covariance. In Foundations of Computer Science (FOCS), 2016 IEEE 57th Annual Symposium on, pp. 665–674. IEEE, 2016.

[2] Ilias Diakonikolas, Gautam Kamath, Daniel M Kane, Jerry Li, Ankur Moitra, and Alistair Stewart. Being robust (in high dimensions) can be practical. arXiv preprint arXiv:1703.00893, 2017.

---

> ### Author Response · Authors · 2018-11-18
> **Response to AnonReviewer2**
>
> Thank you for your comments. We give a response to each of your comments in Cons and Questions.
>
> Cons:
>
> - We agree that we do not have any convergence guarantee. This is indeed an important problem we hope to address in future work. In this work, we focus on the connection between the depth function and GAN. Compared with the existing polynomial-time methods on robust estimation, this framework is more general in developing robust estimation methods for problems other than mean estimation. For example, the problem of robust covariance matrix estimation and robust regression can be studied within the same framework given the connections to regression depth and matrix depth. Another contribution we would like to emphasize is the study on the effect of the discriminator class, which is of its own importance in understanding GAN. For example, we show that a one-layer net does not work for robust mean estimation using JS-GAN. One has to use a two-layer net.
> - We agree with this comment. JS-GAN is the one that we recommend in the paper. TV-GAN is theoretically optimal, but it does not work in practice when the contamination distribution is not close to the true model, and this is reflected in our numerical results. We somehow need TV-GAN to serve as a connection between depth functions and other f-GANs, and we also need to show its numerical results to convince readers that TV-GAN is not a good choice in practice.
> - This is a very important comment. Tukey’s median is attractive not only because it achieves the minimax rate under the contamination model, but also because of the following four properties: 1). It has a clean objective function that allows easy-to-understand extensions to other problems (regression depth and matrix depth). 2). it does not require the knowledge of the contamination proportion \epsilon. 3). it is adaptive to the unknown covariance structure. 4). it is adaptive and optimal for location estimation under general elliptical distributions. These four properties distinguish Tukey’s median from the existing polynomial-time methods in the literature. The 4th property is especially important, which is a fundamental difference between Tukey’s median and the robust mean estimators in the literature. The existing methods estimate the population mean, while Tukey’s median estimates the population median. The two can be different for many multivariate distributions. In particular, for multivariate Cauchy, there is no mean, but Tukey’s median is still able to achieve the minimax rate under the contamination model of estimating the Cauchy location. The proposed estimator JS-GAN is indeed adaptive to the general class of elliptical distributions, and the new theorem will be included in the revised manuscript. In fact, we only need to change the generator class from Gaussian to the class of elliptical distributions. There is no need to change the discriminator class. Our numerical results also show that if the data is generated from heavy elliptical distributions such as Cauchy, JS-GAN works very well, but dimension halving and iterative filtering do not work as well as our method, because these methods are designed only for robust mean estimation, which is for a different purpose. With this revision, JS-GAN also shares the four properties of Tukey’s median, and is computationally much better than Tukey’s median. We agree with you that we use Scheffe set between two Gaussians, which is a half-plane, to derive TV-GAN. However, for JS-GAN, the Scheffe set, which can be regarded as a one-layer neural net, does not work (see discussion after Proposition 3.1). One has to use two-layer neural nets, which is not the Scheffe set between two Gaussians anymore. The overall connection between GAN and depth functions is most clear in a Gaussian framework, but the derived estimator works for general elliptical distributions.
>
> Questions:
>
> - The computational cost is comparable to, but slower than, both dimension halving and iterative filtering. This is because training a two-layer net is a harder optimization problem. The good news is the plot that shows the relation between dimension and computational time is approximately linear, so the method is scalable. Previously, Tukey’s median never works when dimension exceeds 10, but now we can compute JS-GAN, which shares the good properties of Tukey’s median in thousands of dimensions.
> - Yes, we will get the same error rate. This is because for TV(P_1,P_2)<\epsilon, there exist Q_1 and Q_2, such that P_1=P_2-\epsilon Q_1 + \epsilon Q_2. Use this fact, and the proof will go through easily. The new Theorem for elliptical distributions is now proved under strong contamination.
>
> The revised manuscript is uploaded with changes highlighted in red.

---

> > ### Comment · AnonReviewer2 · 2018-11-25
> > **Response to author comments**
> >
> > [edit: forgot to add reference]
> >
> > - "TV-GAN is theoretically optimal, but it does not work in practice when the contamination distribution is not close to the true model"  I find this comment a bit puzzling. It is true that if one could truly optimize the TV- GAN objective the solution would recover the ground truth, but the algorithm presented does not do this, as the TV-GAN algorithm runs some greedy first order method to attempt to approximate this. However, the authors experiments demonstrate that the TV-GAN algorithm does not always converge, and as a result TV-GAN is far from theoretically optimal.
> >
> > - In regards to this comment as well as their response to Reviewer 3, I am a bit confused. I do not see why any of their derivations should hold for anything beyond specifically Gaussian distributions. The authors make a lot of claims about the nice properties of JS-GAN which I do not believe they can support (see my comment at the bottom).
> >
> > - If indeed the algorithm gets the same error rate in the presence of stronger adversaries, then it seems extremely unlikely that the algorithm can be made algorithmic. This is because there are strong SQ lower bounds against getting O(eps) error for estimating the mean of a Gaussian under these stronger adversaries [1], and it is easy to check that the sorts of SGD operations that the GAN methods use fall into this class of algorithms.
> >
> > Generally, it seems to me that the authors are conflating two things, which I think is potentially dangerous. There is the true objective that the authors write down for something such as TV-GAN or JS-GAN, and then there is the actual output of the algorithm based on training via SGD. While I agree that the actual objective has very nice properties (as these are more or less classical statistical objectives), I am very unconvinced that the output of the algorithm has any of these nice properties.
> >
> > [1] Statistical Query Lower Bounds for Robust Estimation of High-dimensional Gaussians and Gaussian Mixtures. I. Diakonikolas, D. Kane, A. Stewart. In FOCS 2017

---

> > > ### Author Response · Authors · 2018-11-27
> > > **(2nd) Response to AnonReviewer2 [PART-1]**
> > >
> > > Thank you again for your additional comments. Before giving a response to your each specific comment, please allow us to clarify our main contributions:
> > >
> > > The paper’s main contribution is to establish a connection between the framework of GANs and the framework of various data depth functions. We believe the connection is important because GAN is important in the deep learning literature and depth functions are important to robust statistics. The connection makes it possible to use tools in one area to solve problems in the other one.
> > >
> > > We understand and fully agree with your assessment for optimal estimation under the contamination model. It is very clear that in terms of provable algorithms that achieve optimal estimation rates, the approach by GANs may suffer from computational intractability in the worst case compared to [1, 2]. Then what is the point to study an old problem using such a new method?
> > >
> > > However, the connection and the framework that we build may provide new insights to both the deep learning area and robust statistics area:
> > >
> > > (i) From the deep learning perspective, it is important to understand how to design the discriminator class to optimally learn a parameter of interest or a generative process using GAN. The paper [3] is such an example for Wasserstein-GAN. The paper [3] does not give a specific algorithm to optimize W-GAN, but even the property of the objective function is already very interesting. Our manuscript is the first to study how to use GAN in the robust setting, and probably even the first to theoretically study how to design discriminator class for JS-GAN towards statistical optimality.
> > >
> > > (ii) From the robust statistics perspective, there are not just Tukey depth, but various depth functions that solve all kinds of robust estimation problem. Our connection might not give a provably polynomial algorithm, but it opens a door for using deep learning tools to optimize the corresponding JS-GAN. Previously, the state of the art of using depth functions only work for data of at most 10 dimensions. Now we can handle 5000 dimensions without a problem. This is an improvement. Even for just Tukey depth, we show that global optimum of Tukey depth or JS-GAN both achieve informationally-theoretic optimality for elliptical distributions. This generality is not met by [1, 2] (To be fair [1, 2] estimate mean, but depth or JS-GAN estimates center, so they are different).
> > >
> > > There is little theory on convergence and computational complexity of GAN, but this does not prevent it from being so useful in learning complex distributions of images. If people use GAN to learn image distributions in practice, why not also use that for robustly learning model parameters? We are not trying to overly manipulate this connection between GAN and depth functions. It does work well in our extensive experiments and the neural net structures are not hard to tune.
> > >
> > > Establishing some convergence theory of GAN will be a fascinating and important topic. However, this is very hard given the current techniques that we have. The state-of-the-art of the area [4,5] analyze the dynamics of the corresponding ODE and the key lies in the construction of a Lyapunov function. This is one of the future projects that we consider, and robust estimation will be a good place to start, given the connection established in this submission.
> > >
> > > Let us clarify that the theorems in submission are all proved for global minimum, and we make this clear in the paper. We did not claim anything of the alternating stochastic gradient algorithm except showing its good numerical results.
> > >
> > > Our response may be a bit aggressive, and we apologize for that. Please understand our enthusiasm behind the work.

---

> > > ### Author Response · Authors · 2018-11-27
> > > **(2nd) Response to AnonReviewer2 [PART-2]**
> > >
> > > (Part 1 of this comment is shown below this one.) Now we give a specific response to each of your comments.
> > >
> > > - "TV-GAN is theoretically optimal, but it does not work in practice when the contamination distribution is not close to the true model" I find this comment a bit puzzling. It is true that if one could truly optimize the TV-GAN objective the solution would recover the ground truth, but the algorithm presented does not do this, as the TV-GAN algorithm runs some greedy first order method to attempt to approximate this. However, the authors experiments demonstrate that the TV-GAN algorithm does not always converge, and as a result, TV-GAN is far from theoretically optimal.
> > >
> > > When we say “theoretical”, we mean for the global optimum. By “in practice”, we mean its numerical performance in the experiments.
> > >
> > > - In regards to this comment as well as their response to Reviewer 3, I am a bit confused. I do not see why any of their derivations should hold for anything beyond specifically Gaussian distributions. The authors make a lot of claims about the nice properties of JS-GAN which I do not believe they can support (see my comment at the bottom).
> > >
> > > These claims are all for the global optimum. For example, Equation (16) in the manuscript.
> > >
> > > - If indeed the algorithm gets the same error rate in the presence of stronger adversaries, then it seems extremely unlikely that the algorithm can be made algorithmic. This is because there are strong SQ lower bounds against getting O(eps) error for estimating the mean of a Gaussian under these stronger adversaries [6], and it is easy to check that the sorts of SGD operations that the GAN methods use fall into this class of algorithms.
> > >
> > > We have realized the paper [6], and your comment is true that this rules out SGD on GAN under the strong contamination model. Our guarantee is only for the global optimum and it does not imply anything on the computational complexity. As we have discussed before, it may be possible to prove convergence of GAN for robust estimation using the techniques in [4,5], and we do think in doing that, further assumptions on the contamination must be necessary.
> > >
> > > [1] Kevin A Lai, Anup B Rao, and Santosh Vempala. Agnostic estimation of mean and covariance. In Foundations of Computer Science (FOCS), 2016 IEEE 57th Annual Symposium on, pp. 665–674. IEEE, 2016.
> > >
> > > [2] Ilias Diakonikolas, Gautam Kamath, Daniel M Kane, Jerry Li, Ankur Moitra, and Alistair Stewart. Being robust (in high dimensions) can be practical. arXiv:1703.00893, 2017.
> > >
> > > [3] Yu Bai, Tengyu Ma, and Andrej Risteski. Approximability of Discriminators Implies Diversity in GANs. arXiv:1806.10586, 2018.
> > >
> > > [4] A. CHERUKURI, B. GHARESIFARD, AND J. CORTES. Saddle-point dynamics: conditions for asymptotic stability of saddle points. SIAM, 2017
> > >
> > > [5] M. Heusel, H. Ramsauer, T. Unterthiner, and B. Nessler. GANs Trained by a Two Time-Scale Update Rule Converge to a Local Nash Equilibrium. NIPS, 2017
> > >
> > > [6] I. Diakonikolas, D. Kane, A. Stewart. Statistical Query Lower Bounds for Robust Estimation of High-dimensional Gaussians and Gaussian Mixtures. In FOCS 2017

---

### Official Review · AnonReviewer3 · 2018-11-09
**Interesting Connection**

**Rating:** 7
**Confidence:** 4

**Review:**

This paper considers the robust estimation problem under Huber’s \epsilon-contamination model. This problem is a hot topic in theoretical statistics and theoretical computer science community in recent 3 years.  From theoretical statistics community, the main approach is through depth functions. Solving the robust estimation problem can be reduced to solving a min-max problem. While the formulation is clean and can achieve the optimal statistical rate, solving the min-max problem is computationally intractable in general. On the other hand, approaches from TCS community are more involved and sometimes cannot achieve the optimal statistical rate (especially for the general distribution).

This paper tries to make the approach from theoretical statistical community computationally tractable. This paper builds an interesting connection between f-GAN and depth functions. Importantly, authors show that by carefully choosing the discriminators neural network architecture and constraining the norms of the weight matrices, the generator achieves the optimal rates. This is an interesting theoretical discovery.

My major question is whether this approach can be used to solve robust estimation problems in more general settings. For example, we want to do robust mean estimation problem and the only assumption on P is it is sub-Gaussian. Is it possible to design a generator-discriminator pair to solve this problem? Theorems in this paper only focus on the Gaussian case.


Overall, I like this paper. This paper provides a new angle toward a classical statistical problem. The computational issue has not been resolved yet. However, given recent progress from optimization in deep learning, it is quite possible that the optimization problem in this paper can be solved (approximately). Therefore, I recommend accepting.

---

> ### Author Response · Authors · 2018-11-18
> **Response to AnonReviewer3**
>
> Thank you for your comment.
>
> Your major question is whether or not the approach can be used to solve robust estimation problems in more general settings. The answer is yes. Even though the submission only considers estimating Gaussian mean, the JS-GAN also works for robust estimation of location vector of a general elliptical distribution. This includes multivariate Cauchy distribution where mean doe not even exist. As a modification, we only need to change the generator class in the JS-GAN from Gaussian to elliptical. There is no need to change the discriminator class. The estimator is minimax optimal under general elliptical family. Our numerical results demonstrate the good performance of the estimator under multivariate Cauchy data.
>
> The revised manuscript is uploaded with changes highlighted in red.

---

> > ### Comment · AnonReviewer3 · 2018-11-27
> > **Thanks for your response**
> >
> > Thanks for your response. The result of elliptical distributions is interesting.

---

### Comment · Area_Chair1 · 2018-12-05
**Clarification about Proposition 2.1**

Proposition 2.1 doesn't type-check for me: The family \tilde{Q} references an unbound variable \eta (as far as I can tell \eta is used as a temporary variable in the definition of Q but does not exist outside of that). Can you please clarify the statement of the proposition?

---

> ### Author Response · Authors · 2018-12-06
> **Response to Area Chair1**
>
> Thank you for the question.
>
> Yes, the statement is a bit confusing. In the formulation $\min_Q\max_{\tilde{Q}}$, notice that $\min_Q$ is before $\max_{\tilde{Q}}$. Thus, the class that we maximize over $\tilde{Q}$ is allowed to depend on $Q$. To be specific, for example in location estimation (Proposition 2.1 (1)), we should regard the definition $\tilde{\mathcal{Q}}_{\eta,r}=\{N(\tilde{\eta},I_p): \|\tilde{\eta}-\eta\|\leq r\}$, as a class that depends on a given $\eta$ and a given $r$. Then, we define $\mathcal{Q}=\{N(\eta,I_p): \eta\in\mathbb{R}^p\}$. In this way, we have $\min_{\eta}\max_{\tilde{\eta}: \|\tilde{\eta}-\eta\leq r\|}$, which means we first maximize $\tilde{\eta}$ near $\eta$, and then minimize over $\eta$. This is exactly equivalent to Equation (20) in Appendix C.

---

### Meta-Review · Area_Chair1 · 2018-12-15
**writing issues outweighed by high conceptual novelty**

**Confidence:** 4
**Recommendation:** Accept (Poster)

**Metareview:**


* Strengths

This paper presents a very interesting connection between GANs and robust estimation in the presence of corrupted training data. The conceptual ideas are novel and can likely be extended in many further directions. I would not be surprised if this opens up a new line of research.

* Weaknesses

The paper is poorly written. Due to disagreement among the authors and my interest in the topic, I read the paper in detail myself. I think it would be difficult for a non-expert to understand the key ideas and I strongly encourage the authors to carefully revise the paper to reach a broader audience and highlight the key insights. Additionally, the experiments are only on toy data.

* Discussion

One of the reviewers was concerned about the lack of efficiency guarantees for the proposed algorithm (indeed, the algorithm requires training GANs which are currently beyond the reach of theory and finicky in practice). That reviewer points to the fact that most papers in the robustness literature are concerned with computational efficiency and is concerned that ignoring this sidesteps one of the key challenges. The reviewer is also concerned about the restriction to parametric or nearly-parametric families (e.g. Gaussians and elliptical distributions). Other reviewers were more positive and did not see these as major issues.

* Decision

In my opinion, the lack of efficiency guarantees is not a huge issue, as the primary contribution of the paper is pointing out a non-obvious conceptual connection between two literatures. The restriction to parametric families is more concerning, but it seems possible this could be removed with further developments. The main reason for accepting the paper (despite concerns about the writing) is the importance of the conceptual connection. I think this connection is likely to lead to a new line of research and would like to get it out there as soon as possible.

* Comments

Despite the accept decision, I again urge the authors to improve the quality of exposition to ensure that a large audience can appreciate the ideas.